# RLSF: REINFORCEMENT LEARNING VIA SYMBOLIC FEEDBACK

## ABSTRACT

Reinforcement Learning with Human Feedback (RLHF) is considered a standard approach to fine-tuning Large Language Models (LLMs) but faces challenges such as unsound reward models, sparse rewards, and difficulties in collecting human preference data, limiting its effectiveness for complex, domain-specific tasks.

We propose Reinforcement Learning via Symbolic Feedback (RLSF), a novel fine-tuning paradigm where reasoning tools (e.g., solvers, provers, algebra systems) serve as the RL environment, providing fine-grained feedback via *poly-sized certificates* (e.g., proofs) that characterize errors in the LLM-generated object with respect to specific correctness criteria. RLSF aims to improve the domain-specific understanding of LLMs more effectively than traditional reward signals. By enabling token-level corrections without requiring differentiable reasoning systems, RLSF addresses key limitations of traditional reward models.

Via extensive evaluations, we show that our RLSF-based fine-tuning of LLMs outperforms traditional approaches on five different applications (that have some associated logical or domain constraints), namely, program synthesis from natural language pseudo-code to programming language (+31.43% in functional correctness for Google's CodeGemma-2b compared to supervised fine-tuning, +17.01% in functional correctness compared to GPT-3.5 – 100× larger), three chemistry tasks (+5.5% exact match for molecule generation, +19.4% exact match for forward synthesis, +33.7% exact match for retrosynthesis, using Meta's Galactica-1.3b, compared to GPT-4 – 1000× larger), and solving the Game of 24 (+25% success rate using Meta's Llama2-7b compared to traditional methods, and +7% success rate compared to GPT-3.5 – 25× larger). A takeaway is that fine-tuning via RLSF enables relatively smaller LLMs to significantly outperform closed-source models that are orders of magnitude larger (e.g., GPT-4).

## 1 INTRODUCTION

In recent years, Large Language Models (LLMs) have had a dramatic impact on many sub-fields of AI (Bommasani et al., 2021). Tasks that seemed impossible just a few years ago, are now routinely solved by LLMs. Examples include language translation (Qin et al., 2024; Min et al., 2023), text-to-image generation (Zhang et al., 2023), coding assistants (Liang et al., 2024), and open-ended text generation (Achiam et al., 2023).

Despite their impressive performance, these models often struggle with tasks that require reasoning or formal domain knowledge (Mao et al., 2024; Stechly et al., 2024; Zhang et al., 2022). This limitation has sparked a growing interest in exploring methods that can better align LLM outputs with logical or domain constraints, particularly through the incorporation of corrective feedback loops between learning and reasoning processes (Ganesh et al., 2022; Kambhampati et al., 2024). For example, Kambhampati et al. (2024) state that "*LLMs cannot plan themselves but can play a variety of constructive roles in solving planning tasks–especially as approximate knowledge sources and candidate plan generators in the so-called LLM-Modulo Frameworks in conjunction with external sound model-based verifiers.*"

The concept of incorporating reasoning tools into machine learning in general, and LLMs in particular, is rooted in the idea of combining the strengths of these two sub-fields of AI (Hitzler et al.,

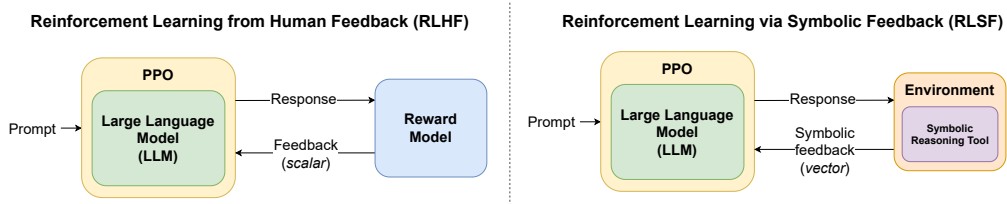

Figure 1: **Contrasting RLHF with RLSF:** The image depicts two distinct fine-tuning paradigms. (Left) RLHF operates within an environment governed by a black-box reward model, typically offering scalar feedback. (Right) By contrast, the environment in RLSF leverages sound symbolic reasoning tools and also provides fine-grained token-level vector (dense) feedback that is, in turn, based on poly-sized certificates produced by these symbolic tools.

2023; Ganesh et al., 2022; Kambhampati et al., 2024; Bouraoui et al., 2019). While LLMs excel at capturing statistical patterns and generating fluent text, they can fail to perform sound reasoning and generate text that is logically coherent. In fact, the logical or code errors introduced by LLMs in the objects they generate can be very subtle, and not immediately obvious upon inspection. This motivates the need to use reasoning and verification tools at different stages of an LLM's life cycle (from data curation, training, fine-tuning, and inference). For instance, using program analysis tools during inference (Agrawal et al., 2024), and integrating solvers into neural network layers (Wang et al., 2024b) or during gradient descent (Ashok et al., 2022) have shown promising results in terms of faster convergence and efficient data utilization.

By contrast to LLMs, symbolic reasoning systems, such as theorem provers and constraint solvers [1], are adept at handling sound logical reasoning, perform symbolic mathematical operations and maintain coherence, but they do not seem to possess the impressive generative capabilities of LLMs. By integrating these two approaches, a hybrid system can be created that can leverage the strengths of both paradigms, potentially leading to more robust and capable AI systems.

One popular approach to fine-tuning LLMs is Reinforcement Learning from Human Feedback (RLHF) (Ouyang et al., 2022; Stiennon et al., 2020), which relies on manually collecting correct/incorrect cases and creating an approximate (possibly unsound) black-box reward model. Such an approach can be expensive, error-prone, and may not accurately capture the nuances of a reasoning task. Moreover, the reward signal thus generated can be sparse and scalar in nature. Such sparse reward signals can fall short of fully capturing those aspects of an LLM-generated object that may be incorrect with respect to a well-defined specification (or inconsistent with domain knowledge).

To address these challenges, we propose a new fine-tuning paradigm we refer to as Reinforcement Learning via Symbolic Feedback (RLSF) that is designed to improve the performance of LLMs compared to traditional methods in complex reasoning tasks. Figure 1 highlights the differences between RLHF and RLSF. In the RLSF setting, the LLM is considered as the RL agent to be fine-tuned, while the environment is allowed access to reasoning tools, that can generate poly-sized certificates of their analyses.

In the forward direction (LLM to environment), formal objects generated (such as programs, proofs, molecules, or theories) by the LLM are fed to the RLSF environment, which leverages reasoning tools to perform analysis or verification of such objects against specifications or domain knowledge. In the reverse direction (environment to LLM), any certificate (that identifies what is wrong with the LLM-generated object) produced by symbolic tools is used as part of a reward function to provide corrective feedback to the LLM. Leveraging symbolic tools to generate poly-sized certificates [2], which provide sound, fine-grained (token-level) feedback to LLMs, eliminates the need for manual

---

[1]We define the term symbolic reasoning systems broadly to include solvers, provers, computer algebra systems, program analysis tools, knowledge bases, and simulators. The only requirement is that they analyze/solve inputs that are formally defined, and can produce poly-sized certificates of their analysis.

[2]Examples of poly-sized certificates include unsatisfiability proofs, compiler feedback, equivalence testing, etc. Our approach is not limited to any one type of symbolic tool, as long as these certificates can be converted into appropriate rewards. It is possible that the problem addressed by these symbolic tools is NP-hard, and therefore, in general, we cannot always expect certificates that are polynomial in the input size. Having said that, many of these tools, such as compilers, computer algebra systems, or solvers, work well in practice.

preference data collection and addresses limitations of traditional reward models. Moreover, our RLSF approach does not require the reasoning systems to be differentiable, unlike traditional neuro-symbolic RL approaches (Hitzler et al., 2023), adding to its versatility.

**Contributions.**

- **Reinforcement Learning with Symbolic Feedback:** We present the RLSF paradigm and evaluate the effectiveness of RLSF-based fine-tuning methodologies on LLMs across five distinct tasks that present unique challenges regarding their domain-specific understanding.

- **Translation of natural language pseudo-code to C++ code:** Our results show a significant improvement over the supervised fine-tuning (SFT) approach for widely-used code LLMs (such as `stable-code-instruct-3b` (Pinnaparaju et al., 2024), `deepseek-coder-1.3b-base` (Guo et al., 2024) and Google's `code-gemma-2b` (CodeGemma-Team, 2024)). For example, our RLSF-tuned `code-gemma-2b` shows an improvement of +52.64% in compilation accuracy and +31.43% in functional correctness accuracy over SFT and also achieves superior results compared to GPT-3.5 ($\sim$100$\times$ larger) (Achiam et al., 2023). (Section 4)

- **Three chemistry tasks (molecule generation, forward synthesis, retrosynthesis):** For the three chemistry tasks, `mistral-7b-v0.1` (Jiang et al., 2023) from Mistral AI and `galactica-1.3b` (Taylor et al., 2022) from Meta AI show an improvement upto 13% in exact match and 58% in validity compared to traditional approaches, and upto 33% improvement in exact match compared to GPT-4 ($\sim$1000$\times$ larger). (Section 5)

- **Game of 24:** We observe significant improvement using RLSF on popular LLMs – Google's `gemma-2b-it` (Gemma-Team et al., 2024) and Meta's `llama2-7b-chat` (Touvron et al., 2023) with +15% and +25% success respectively, compared to traditional methods. Notably, post RLSF fine-tuning, `llama2-7b-chat` also outperforms (+7%) GPT-3.5 ($\sim$25$\times$ larger). This underscores the importance of RLSF fine-tuning that facilitates relatively smaller LLMs to significantly outperform models, such as ChatGPT, which are orders of magnitude larger. (Section 6)

## 2 RELATED WORK

The idea of integrating symbolic feedback into reinforcement learning for RLSF has philosophical origins in the Learning Rate Based (LRB) branching heuristic for SAT solvers (Liang et al., 2016). In their work, they model the variable selection problem as an online multi-armed bandit, a special case of reinforcement learning, to learn branching variables such that the learning rate of the solver is maximized. LRB uses a structured feedback signal based on the solver's performance and conflict analysis, guiding the optimization of variable selection. This is one of the key inspirations behind the RLSF approach, where we provide fine-grained feedback to LLMs via poly-sized certificates generated by symbolic reasoning tools.

**Neurosymbolic Reinforcement Learning (NRL).** We refer the reader to the Compendium of Neurosymbolic AI for a rich and recent literature survey on combinations of learning and reasoning systems (Hitzler et al., 2023). There has been considerable work in NRL (Acharya et al., 2023). However, all the NRL work that we are aware of focuses on combining symbolic reasoning tools with the RL agent, often requiring reasoning systems to be differentiable and also limiting the agent's ability to handle large symbolic representations. In contrast, RLSF integrates symbolic tools with the environment and eliminates the need for differentiability. Further, in the RLSF paradigm, a key innovation is that we leverage the poly-sized certificates generated by the environment's symbolic tools to provide a fine-grained reward signal to the RL agent. This allows for a modular framework with more expressive and interpretable feedback during training.

**Combinations of LLMs with Symbolic Tools.** Previous research has explored the integration of program analysis tools during inference (Agrawal et al., 2024). While existing efforts have often relied on binary or scalar signals for utilizing symbolic feedback during training or fine-tuning (Chen et al., 2020; Jana et al., 2023; Jha et al., 2024; Zou et al., 2024), the RLSF paradigm leverages richer token-level (dense) symbolic feedback, thus enabling significantly improved fine-tuning performance as it enables more detailed and fine-grained correction, pinpointing specific areas in the output that need improvement. This allows the model to learn more effectively, as it receives clearer,

---

**Algorithm 1** Reinforcement Learning via Symbolic Feedback (RLSF)

---

**Input:** Number of epochs $N_{epochs}$, pre-trained model $Model$, symbolic reasoning tool $SymbolicReasoner$, reward function $RewardFunc$, prompt dataset $D$
**Output:** Fine-tuned model $Model'$
  **for** $N_{epochs}$ **do**
    **for** $batch_i$ in $D$ **do**
      $response_i \sim Model(batch_i)$                       ▷ Generate a response
      $cert_i \leftarrow SymbolicReasoner(batch_i, response_i)$       ▷ Compute the certificate
      $vector_i \leftarrow RewardFunc(cert_i)$           ▷ Compute the vector feedback
      $Model' \leftarrow ppo\_step(Model, batch_i, response_i, cert_i)$    ▷ Update model using PPO
      $Model \leftarrow Model'$
    **end for**
  **end for**

---

more precise guidance. Kambhampati et al. (2024) introduced the concept of LLM-Modulo Frameworks, which proposes using external symbolic reasoning tools during inference. In contrast, our work focuses on the fine-tuning phase and how external symbolic reasoning tools can be effectively incorporated into the RLSF paradigm to guide the model during fine-tuning.

**LLM-based Feedback.** Another line of work that has explored using LLMs as part of a corrective feedback loop (Shinn et al., 2024; Madaan et al., 2024; Paul et al., 2023; Kim et al., 2024; Chen et al., 2023a) where the feedback is usually provided as in-context (not RL), under the assumption that LLMs are better at verification compared to generation. However, recent studies have challenged this assumption, suggesting that LLMs may not be as effective at verification/self-critiquing tasks as previously thought (Mao et al., 2024; Stechly et al., 2024; Valmeekam et al., 2023; Huang et al., 2023; Stechly et al., 2023). By contrast, in our work, we use sound symbolic systems to provide corrective feedback and do so via an RL approach.

## 3   REINFORCEMENT LEARNING VIA SYMBOLIC FEEDBACK (RLSF)

In this section, we introduce the Reinforcement Learning via Symbolic Feedback (RLSF) algorithm. The algorithm incorporates fine-grained, token-level feedback generated by reasoning or domain knowledge tools, thereby addressing the limitations of traditional reward-based methods.

The RLSF algorithm, outlined in Algorithm 1, fine-tunes a pre-trained language model $Model$ using reinforcement learning (RL) with the help of the fine-grained certificate provided by a symbolic reasoning tool $SymbolicReasoner$. The framework leverages a reward function ($RewardFunc$) to compute vector (token-level) feedback ($vector_i$) based on the certificate generated by the symbolic reasoning tool. This feedback aligns with the shape of the response generated by the language model, facilitating fine-grained adjustments during fine-tuning. The algorithm operates over a specified number of epochs $N_{epochs}$, iterating through a dataset $D$.

**Inputs and Outputs.** The algorithm takes as input the pre-trained model (RL agent to be fine-tuned) $Model$, the symbolic reasoning tool $SymbolicReasoner$ and the reward function $RewardFunc$ (to be used as the RL environment), and the dataset $D$ that consists of input prompts for $Model$. The output is a fine-tuned model $Model'$. The algorithm performs the following steps:

**Epoch Iteration.** For each epoch from 1 to $N_{epochs}$, the algorithm iterates through $D$.

**Batch Processing.** For each batch $batch_i$ in the dataset $D$, the algorithm performs the following:
- Response generation ($response_i$): The $Model$ generates a response $response_i$ given the input prompt $batch_i$.
- Certificate computation ($cert_i$): The symbolic reasoning tool $SymbolicReasoner$ computes a certificate $cert_i$ corresponding to the response $response_i$. The certificate includes fine-grained error/non-error messages extracted from the symbolic analysis of the prompt-response pair.
- Token-level (dense) feedback computation ($vector_i$): The reward function ($RewardFunc$) calculates a vector feedback ($vector_i$) based on the certificate $cert_i$. The reward function processes this certificate to generate token-level vector feedback. This feedback provides detailed guidance to the language model during fine-tuning, facilitating precise adjustments. The vector feedback

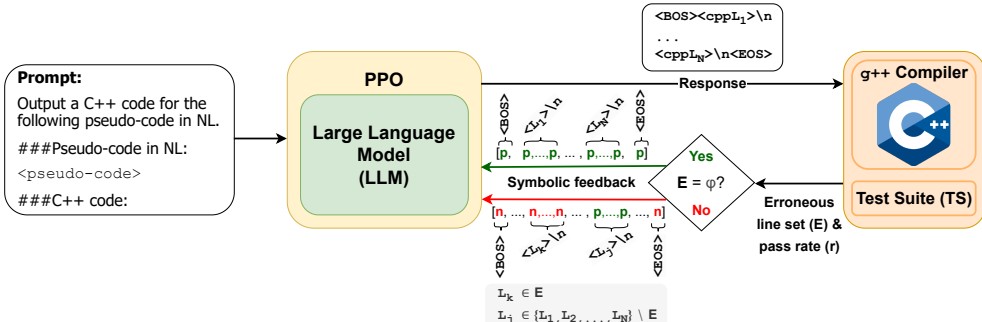

Figure 2: **RLSF for translation of NL pseudo-code to code:** Given the generated C++ code (with N lines), the symbolic environment uses the g++ compiler to detect erroneous lines ($E$) and compute the pass rate $r$ from the test suite, providing fine-grained symbolic feedback for fine-tuning the LLM.

has the same shape as the response tokens (computed using the tokenizer provided by *Model*). Figures 2, 3 and 4 give a concrete example of such a vector feedback for the different tasks.

- Model update ($Model'$): The $Model$ is updated to $Model'$ using the Proximal Policy Optimization (PPO) algorithm, using the input prompt $batch_i$, response $response_i$, and certificate $cert_i$.

**Generalizing RLSF to different reasoning tasks.** Our RLSF paradigm can be applied to any reasoning task, provided that: *(a)* the final output is in a formal language, and *(b)* a $SymbolicReasoner$ can furnish segment-wise feedback based on a chosen delimiter. In Algorithm 1, we fine-tune an LLM for a specific reasoning task where the output is expressed in a formal language. Since each $response_i$ must conform to a formal language $F$, logical delimiters are used to segment $response_i$ into lines (using \n), words (using spaces), characters, or parser-tokens based on the grammar of $F$. When validated by the $SymbolicReasoner$, $cert_i$ verifies each segment. Then $RewardFunc$ maps the segment-level certificate to $vector_i$, providing token-level feedback for $response_i$. In Sections 4, 5 and 6, we demonstrate the RLSF paradigm across five reasoning tasks from different domains. The implementation details of the RLSF algorithm are described in Section D.

# 4 REASONING TASK A: NATURAL LANGUAGE PSEUDO-CODE TO C++ CODE

Automated code synthesis from natural language (NL) descriptions is a crucial task in software engineering that has garnered considerable attention. Recent efforts, such as Ugare et al. (2024), have explored using LLMs for this. We evaluate how our RLSF paradigm improves LLM-based translation of an NL pseudo-code into a C++ code. To be correct, the code must be both (a) *syntactically correct* per the g++ compiler and (b) *functionally correct* (or equivalent [3]) w.r.t. a test suite.

## 4.1 BENCHMARK AND RLSF SETUP

For training and evaluating the LLM, we utilize the SPoC dataset (Kulal et al., 2019), which includes 16,326 accepted C++ solutions for 677 problems from a competitive programming website (Codeforces, 2023). On average, each program has 14.7 lines, ranging in $[1, 457]$. They employed 59 crowd-workers from Amazon Mechanical Turk to write NL pseudo-code for each line of C++ program, ensuring a line-level granularity of textual description. Each problem is accompanied by a test suite of multiple test cases, curated by the problem-setter. We evaluate on the TESTP partition of SPoC, consisting of 158 problems ($\sim$23.34%) and 1,778 pairs of pseudo-code and C++ code. The remaining 14,548 pairs from 519 problems ($\sim$76.66%) are used for training the LLMs. Please refer to Section A.1 in the Appendix for additional implementation details.

Refer to Figure 2 for an overview of the RLSF setup for fine-tuning an LLM to translate pseudo-code into code. In the supervised learning setup, we assume a training dataset $\mathcal{D} = \{(pc_i, c_i, TS_i)\}$, con-

---

[3]Note that in general the task of determining whether a C++ code is functionally equivalent to pseudo-code (even when it is specified in formal logic) is undecidable. Hence, we consider a weaker functional equivalence property, namely, that a generated C++ code is deemed functionally correct if it passes all test cases corresponding to a given pseudo-code.

sisting of pseudo-code $pc_i$, gold-standard code $c_i$, and test-suite $TS_i$. During fine-tuning, the LLM receives $pc_i$ as input (with prompt in Figure 2) and generates code $\hat{c}_i$, which may contain syntactic errors. This code is passed through the compiler (here, g++) to identify lines with errors ($E$). If $\hat{c}_i$ compiles, it is tested on $TS_i$ to compute the pass rate ($r \in [0, 1]$). If it does not compile, $r = 0$. Our token-level fine-grained feedback assigns a high reward $p = 1 + r$ to tokens in syntactically correct lines (positive case) and a low reward $n = 0$ to tokens in erroneous lines (negative case).

Our setup features a gradual progression of rewards and fine-grained feedback. Tokens in erroneous lines receive a reward of 0, while those in syntactically correct lines of non-compiling code get a reward of 1 (since $p = 1 + r$ and $r = 0$). Tokens in a compiling code receive rewards $p \in [1, 2]$, based on the value of $r$. The special tokens `<BOS>` and `<EOS>` get a reward of $p$ if $\hat{c}_i$ compiles, and $n$ otherwise. This token-level feedback fine-tunes the LLM through RL using PPO (Algorithm 1).

## 4.2 Evaluation Metrics

We use three metrics to evaluate pseudo-code to C++ code translation with LLMs. CompAcc and FCorrAcc provide more nuanced insights, while Pass@1 is a stricter pass/fail metric. In Section 4.4, we present performance using all three metrics, focusing on the first two for comparison of methods.

**Compilation Accuracy (CompAcc).** The percentage of generated C++ codes that are syntactically correct, indicating that it compiles without errors using a g++ compiler.

**Functional Correctness Accuracy (FCorrAcc).** The percentage of test cases demonstrating the expected input-output behavior for each generated C++ code, averaged across all the generated codes. If a generated code contains syntactical errors, FCorrAcc is zero for that code.

**Pass@1.** The percentage of generated C++ codes generated on the first attempt by the LLM that is syntactically correct and passes all test cases in the problem's test suite, with each code evaluated as *true* if it passes all tests and *false* otherwise.

## 4.3 Comparative Models Used

We perform RLSF fine-tuning on three recent open-source code LLMs that we obtain from HuggingFace (2024): `code-gemma-2b` from Google, `stable-code-instruct-3b` from StabilityAI, and `deepseek-coder-1.3b-base` from DeepSeekAI. We also compare results with the closed-source `gpt-3.5-turbo-0301` model i.e., `GPT-3.5` (ChatGPT), that we access via the API of OpenAI (2023). For reproducibility, temperature and top_p are set to 0. The three open-source models have 2 billion, 2.7 billion, and 1.3 billion parameters, respectively. In comparison, `GPT-3.5` is rumored to have around 175 billion parameters – about 100 times more.

## 4.4 Results and Ablation Study

As shown in Table 1, we first evaluate the LLMs in a **zero-shot setting**. `GPT-3.5` generates 29.13% compilable C++ code, with 24.29% of test cases showing correct input-output behavior. In contrast, none of the open-source models (`code-gemma-2b`, `stable-code-instruct-3b`, and `deepseek-coder-1.3b-base`) produce a single compilable or functionally correct code. Next, we train the open-source models by **supervised fine-tuning (SFT)** to minimize the cross-entropy loss between the generated and gold-standard C++ code. After SFT, `code-gemma-2b` and `stable-code-instruct-3b` achieve about 12% CompAcc and 10% FCorrAcc, while `deepseek-coder-1.3b-base` reaches only around 2% in both metrics.

We further fine-tune the SFT-tuned LLMs through **RL with Boolean feedback**, where models receive a binary scalar reward (0 or 1) based on whether the generated code compiles – a common approach in various recent code generation techniques (Wang et al., 2022; Shojaee et al., 2023; Dou et al., 2024). This improves CompAcc by +43.58%, +36.39%, and +17.78%, and FCorrAcc by +14.84%, +0.13%, and +6.17% for `code-gemma-2b`, `stable-code-instruct-3b`, and `deepseek-coder-1.3b-base`, respectively. Conversely, we use the proposed scheme of **RLSF with token-level feedback** to fine-tune the SFT-tuned LLMs. This yields even better results, improving CompAcc by +52.64%, +42.23%, and +36.73%, and FCorrAcc by +31.43%, +7.66%, and +23.99% for the same models compared to SFT. We believe RLSF's superior performance is due to its token-level feedback. It provides more granular rewards based on line-level

Table 1: **Natural Language Pseudo-code to Code Translation Results:** Performance comparison of fine-tuning methodologies over different LLMs

| LLM | Configuration | CompAcc (%) | FCorrAcc (%) | Pass@1 (%) |
|---|---|---|---|---|
| `GPT-3.5` (Achiam et al., 2023) | Zero-shot (no fine-tuning) | **29.13** | **24.29** | **20.98** |
| `code-gemma-2b` (CodeGemma-Team, 2024) | Zero-shot (no fine-tuning) | 0.00 | 0.00 | 0.00 |
| | SFT (with cross-entropy loss) | 11.31 | 9.87 | 9.00 |
| | SFT + RL (with Boolean scalar f/b) | 54.89 | 24.71 | 12.77 |
| | SFT + *RLSF (with token-level f/b)* | **63.95** | **41.30** | **28.80** |
| `stable-code-instruct-3b` (Pinnaparaju et al., 2024) | Zero-shot (no fine-tuning) | 0.00 | 0.00 | 0.00 |
| | SFT (with cross-entropy loss) | 12.04 | 10.78 | 9.96 |
| | SFT + RL (with Boolean scalar f/b) | 48.43 | 10.91 | 9.22 |
| | SFT + *RLSF (with token-level f/b)* | **54.27** | **18.44** | **16.09** |
| `deepseek-coder-1.3b-base` (Guo et al., 2024) | Zero-shot (no fine-tuning) | 0.00 | 0.00 | 0.00 |
| | SFT (with cross-entropy loss) | 2.19 | 1.90 | 1.63 |
| | SFT + RL (with Boolean scalar f/b) | 19.97 | 8.07 | 3.88 |
| | SFT + *RLSF (with token-level f/b)* | **38.92** | **25.89** | **14.51** |

syntactical correctness and overall test case pass rate, unlike Boolean feedback which only indicates if code compiles. As a result, RLSF-tuned models outperform those trained with Boolean feedback by $+9.06\%$, $+5.84\%$, and $+18.95\%$ in CompAcc, and $+16.59\%$, $+7.53\%$, and $+17.82\%$ in FCorrAcc. They also surpass `GPT-3.5` (ChatGPT), despite using 100 times fewer parameters. Additional results comparing Boolean compiler and success feedback in RL-based fine-tuning can be found in Section A.2 of the Appendix.

# 5 REASONING TASKS B: CHEMISTRY (MOLECULE GENERATION, FORWARD SYNTHESIS AND RETROSYNTHESIS)

Chemistry tasks, such as molecule generation, forward synthesis, and retrosynthesis, are critical benchmarks for assessing the capabilities of LLMs in real-world scientific applications (Schwaller et al., 2019; Zhong et al., 2022; Zhou et al., 2023; Chen et al., 2023b). These tasks require models to navigate the intricate rules of chemical structures and reactions, making them excellent challenges to evaluate domain-specific understanding. Leveraging LLMs in chemistry has the potential to significantly accelerate fields like drug discovery and materials science, enabling faster innovation and discovery (Guan & Wang, 2024; Yu et al., 2024b; Bhattacharya et al., 2024; Jiang et al., 2024).

## 5.1 BENCHMARK AND RLSF SETUP

We focus on three tasks, namely, **Molecular Generation (MG)**, **Forward Synthesis (FS)**, and **Retrosynthesis (RS)**. MG involves generating molecules given a description in natural language. FS involves predicting the product of a chemical reaction given specific reactants and reagents. RS focuses on identifying the reactants necessary to produce a specific target product. All output molecules are generated in the SMILES format (Weininger, 1988), a widely used method for encoding molecular structures as a sequence of symbols, making it a popular choice for LLM-based chemistry models (Yu et al., 2024a). Section B.1 gives more details about the different tasks. USPTO-full dataset (Lowe, 2017) is a commonly used for FS and RS tasks, while ChEBI-20 and Mol-Instructions datasets (Edwards et al., 2022; Fang et al., 2023) are used for the MG task. We use a high-quality version of these datasets from LlaSMol (Yu et al., 2024a), ensuring the removal of chemically invalid SMILES and inaccurate information. We focus on a subset of the dataset where multiple outputs are possible for a given input, as this poses a significant challenge for LLMs. In such cases, the model must better capture the syntax and semantics of a given task, rather than relying on simple input-output mapping through supervised fine-tuning (SFT). This setup pushes the model beyond memorization, testing its ability to generalize, making it a key evaluation of its capabilities. We ensure that same input samples are not shared across dataset splits for the same and also across the three tasks. More details can be found in Section B.2.

For the RLSF feedback, we handle both syntax and semantic errors. In our chemical modeling tasks, we use RDKit (RDKit, 2023), a widely adopted cheminformatics toolkit, as the symbolic feedback generator. RDKit analyzes model outputs, such as SMILES strings, and generates error logs that we transform into token-level rewards whenever applicable. For instance, consider the model-generated SMILES string `C(C)(C)N(C)(C)(C)`. RDKit identifies an issue with the nitrogen atom, flagging it for violating valence rules. Specifically, RDKit's parser outputs the error: *"Explicit valence for atom # 3 N, 4, is greater than permitted."* This indicates that the nitrogen atom (`N`, indexed as

Table 2: **Chemistry Tasks Results:** Performance comparison over different LLMs for three chemistry tasks - Molecule Generation (MG), Forward synthesis (FS), and Retrosynthesis (RS)

| LLM | Configuration | Task & Metric | | | | | | | | |
|---|---|---|---|---|---|---|---|---|---|---|
| | | Molecule Generation (MG) | | | Forward synthesis (FS) | | | Retrosynthesis (RS) | | |
| | | EM (%) | FTS (%) | Valid (%) | EM (%) | FTS (%) | Valid (%) | EM (%) | FTS (%) | Valid (%) |
| GPT-4 (Achiam et al., 2023) | Zero-shot | **3.0** | **35.6** | **90.0** | **0.4** | **37.5** | **93.1** | **0.8** | **31.7** | **88.2** |
| mistral-7b (Jiang et al., 2023) | Zero-shot | 0.0 | 0.0 | 0.0 | 0.0 | 49.8 | 14.4 | 0.0 | 46.4 | 13.5 |
| | SFT | 1.1 | 25.4 | 51.9 | 7.2 | 54.2 | 93.0 | 23.3 | 62.1 | 98.0 |
| | SFT + RL w/ scalar f/b | 4.0 | 31.2 | 62.8 | 10.4 | 56.1 | 94.2 | 28.5 | 64.6 | 98.0 |
| | *SFT + RLSF (token-level f/b)* | **13.6** | **45.1** | **89.1** | **21.3** | **59.7** | **98.3** | **32.1** | **65.8** | **99.1** |
| galactica-1.3b (Taylor et al., 2022) | Zero-shot | 0.0 | 0.0 | 0.0 | 0.0 | 44.8 | 2.2 | 0.0 | 43.2 | 0.2 |
| | SFT | 0.5 | 10.1 | 23.1 | 8.0 | 56.3 | 97.7 | 22.3 | 59.2 | 96.3 |
| | SFT + RL w/ scalar f/b | 2.7 | 27.5 | 75.4 | 11.7 | 56.8 | **98.8** | 26.1 | 62.7 | 99.0 |
| | *SFT + RLSF (token-level f/b)* | **8.5** | **38.2** | **81.4** | **19.8** | **60.3** | **99.8** | **34.5** | **64.4** | **99.5** |

atom 3) has a valence of 4, which exceeds the permissible limit for nitrogen. To correct this error, two possible modifications can be applied: either adding a positive charge to nitrogen to form a quaternary ammonium ion (e.g., `C(C)(C)N+(C)(C)`) or reducing the number of bonds to nitrogen by trimming one methyl group, resulting in a valid structure like `C(C)(C)N(C)(C)`. In this case, erroneous tokens include `N` (penalized for exceeding valence limits) and the parenthetical `C` groups following `N` (penalized as contributors to the invalid valence state).

In addition to valence issues, RDKit helps detect a wide range of errors, including hallucinated (non-existent) atoms, syntax errors (e.g., invalid characters, unbalanced parentheses, or unclosed rings), and semantic violations, such as the introduction of elements that violate conservation laws. RDKit's consistent error format allows for straightforward parsing and categorization of issues, enabling targeted feedback. For example, in FS, we ensure adherence to the first law of chemistry by penalizing outputs that introduce atoms not present in the reactants. If the reactants contain hydrogen and carbon but the product introduces nitrogen, this discrepancy triggers a penalty. Similarly, for molecule generation, we verify the presence of specified functional groups and penalize outputs that omit them (Figure 3). The token-level feedback mechanism ensures that the model learns precise corrections by addressing both syntax and semantic errors effectively. By integrating detailed checks and aligning feedback with domain-specific requirements, our approach significantly enhances the model's ability to generate accurate, chemically valid outputs.

## 5.2 EVALUATION METRICS

Three main metrics are used to assess performance on the FS, RS and MG tasks: **Exact Match** (EM), **Fingerprint Tanimoto Similarity** (FTS), and **Validity** (Valid). EM measures the proportion of predicted results that exactly match the gold standards. FTS quantifies structural similarities between molecules using Tanimoto similarities of their Morgan fingerprints (Morgan, 1965). Validity assesses the ratio of valid predictions following SMILES grammar and the chemical valence rules (Yu et al., 2024a). These metrics are commonly used in the field of cheminformatics and molecular prediction (Schwaller et al., 2020; Chen et al., 2023b; Yu et al., 2024a).

## 5.3 COMPARATIVE MODELS USED

We conduct RLSF fine-tuning on two recent open-source code LLMs that we obtain from HuggingFace (2024): `mistral-7b-v0.1` (Jiang et al., 2023) from Mistral AI and `galactica-1.3b` (Taylor et al., 2022) from Meta AI. We also compare results with the closed-source `gpt-4-turbo` model i.e., `GPT-4` (ChatGPT), that we access via the API of OpenAI (2023). The two open-source models have 7 billion and 1.3 billion parameters, respectively. In comparison, `GPT-4` is rumored to have around 1.76 trillion parameters – about 1000 times more.

## 5.4 RESULTS AND ABLATION STUDY

As shown in Table 2, we first evaluated the LLMs in a zero-shot setting. GPT-4 generated valid SMILES for 90% of the MG inputs but achieved only 3% exact match (EM) with the gold outputs. For the FS task, GPT-4 produced 93.1% valid SMILES and only 0.4% EM, while in the RS task, it generated 88.2% valid SMILES with just 0.8% EM. In contrast, none of the open-source models achieved any EM in the zero-shot setting.

Next, we trained two open-source models using **supervised fine-tuning (SFT)**, which improved their performance in terms of EM, Fingerprint-based Tanimoto Similarity (FTS), and valid SMILES generation. Following SFT, we applied **reinforcement learning (RL) with Boolean feedback**, where the model received a binary reward (0 or 1) based on whether the generated output adhered to task specifications as described in Section 5.1. This further improved performance: for `mistral-7b-v0.1`, we observed a +2.9% increase in EM and a +10.9% boost in validity for the MG task, +3.2% EM and +1.2% validity for FS, and +5.2% EM with no change in validity for RS. Similarly, for `galactica-1.3b`, we recorded a +2.2% increase in EM and +52.3% improvement in validity for MG, +3.7% EM and +1.1% validity for FS, and +3.8% EM with +2.7% validity for RS, compared to the SFT-tuned models.

Moreover, our **RLSF with token-level feedback** approach on the SFT-tuned LLMs, achieved the best results in terms of EM, FTS, and validity. For example, RLSF on `galactica-1.3b` resulted in +8% EM and +58.3% validity for MG, +11.8% EM and +2.1% validity for FS, and +12.2% EM and +3.2% validity for RS, compared to SFT-tuned models. We attribute this superior performance to token-level feedback, which allows for more fine-grained corrections than Boolean feedback, leading to significant improvements. Note that RLSF-tuned models outperformed GPT-4 despite using 1000 times fewer parameters, as shown in Table 2.

# 6 Reasoning Task C: Game of 24 using Tree of Thoughts (ToT)

The Game of 24 is a well-known benchmark that involves basic arithmetic operations such as addition, subtraction, multiplication, and division aimed at testing the mathematical capabilities of LLMs. Briefly, the idea behind the Game of 24 is as follows: given 4 numbers and basic arithmetic operations, obtain the target number 24. We refer the reader to the paper on Tree of Thoughts (ToT) by Yao et al. (2024) and Section C for more details.

## 6.1 Benchmark and RLSF setup

Similar to Yao et al. (2024), we collect data from `4nums.com`, a website hosting mathematical games, selecting 1,362 games sorted by human solving time from easy to hard. For testing, we use the same games as (Yao et al., 2024), indexed 901-1,000. Success in each task is defined as producing a valid equation that equals 24 while using each input number exactly once. For the RLSF fine-tuning phase, we use games indexed 800-900. Fine-tuning occurs during the "propose prompt" steps (using prompt styles from Yao et al. (2024)), as shown in Figure 4. Pairs of "propose prompts" and LLM responses are periodically evaluated using SymPy (Meurer et al., 2017) to identify syntax and semantic errors. This feedback is used to fine-tune the LLM with Proximal Policy Optimization (PPO) following Algorithm 1. For more details on different error categories and converting symbolic feedback into fine-grained signals for fine-tuning, refer to Section C.2.

## 6.2 Comparative Methods and Models Used

We incorporate the benchmarks previously used by Yao et al. (2024), i.e., standard Input-Output (IO) prompting, Chain-of-Thought (CoT) prompting, and ToT prompting. IO prompting uses five in-context examples, while CoT prompting includes three intermediate equations for these in-context examples. Both IO and CoT prompting are sampled 100 times per game for average performance assessment. To showcase the improvement due to fine-grained token-level feedback, we perform an ablation study where we compare the binary (scalar) and token-level versions of feedback for the RL fine-tuning. After RL fine-tuning, we evaluate the performance on the test set using ToT with the updated LLM. We perform RLSF fine-tuning on two popular smaller open-source LLM models (`gemma-2b-it` (Gemma-Team et al., 2024) and `llama2-7b-chat` (Touvron et al., 2023)) that we obtain from Huggingface (HuggingFace, 2024) and also compare against two closed-source models GPT-3.5 and GPT-4 (Achiam et al., 2023).

## 6.3 Results

We conduct a comparative analysis of different methods applied across various LLMs to tackle the Game of 24 task (Table 3). As observed by (Yao et al., 2024), the Tree of Thoughts (ToT) prompt

Table 3: **Game of 24 Results:** Performance comparison of methods over different LLMs

| LLM | Configuration | Success |
|---|---|---|
| GPT-3.5 (Achiam et al., 2023) | IO prompt | 6% |
| | CoT prompt | 3% |
| | ToT prompt | **19%** |
| GPT-4 (Achiam et al., 2023) | IO prompt | 7.3% |
| | CoT prompt | 4% |
| | ToT prompt | **69%** |
| gemma-2b-it (Gemma-Team et al., 2024) | IO prompt | 1% |
| | CoT prompt | 3% |
| | ToT prompt | 2% |
| | ToT after RL with Boolean scalar f/b | 1% |
| | *ToT after RLSF with token-level f/b* | **17%** |
| llama2-7b-chat (Touvron et al., 2023) | IO prompt | 5% |
| | CoT prompt | 1% |
| | ToT prompt | 1% |
| | ToT after RL with Boolean scalar f/b | 1% |
| | *ToT after RLSF with token-level f/b* | **26%** |

method emerges as the most successful approach for both closed-source models GPT-3.5 and GPT-4, achieving success rates of 19% and 69%, respectively. This performance surpasses that of both the IO and CoT prompt methods. However, gemma-2b-it and llama2-7b-chat exhibit **poor performance across all the three prompting methods**.

We explore the use of Boolean scalar feedback for RL fine-tuning, where the Computer Algebra System (CAS) provides binary feedback based on the correctness of the generated response. However, we observe either degradation or maintenance of performance levels with this feedback mechanism (Table 3). Consequently, we transition to a token-level feedback approach using RLSF, where the symbolic environment provides token-level feedback (Figure 4), resulting in a significant improvement in performance. Specifically, after employing Reinforcement Learning via Symbolic Feedback (RLSF) fine-tuning, gemma-2b-it demonstrates a 15% increase in success rate, while llama2-7b-chat exhibits a 25% improvement in success rate compared to ToT prompting prior to RLSF fine-tuning. Notably, the 7-billion-parameter llama2-7b-chat outperforms (+7%) the 175-billion-parameter GPT-3.5 model, underscoring the effectiveness of RLSF in enhancing model performance. However, GPT-4 demonstrates the best performance across all methods. We attribute this to its ultra-large-scale pre-training and architecture advancements. Looking ahead, future investigations can explore the application of RLSF on larger open-source LLMs.

# 7 CONCLUSIONS, LIMITATIONS, AND FUTURE WORK

In this paper, we introduced RLSF, a fine-tuning paradigm that incorporates RL-based symbolic feedback into the fine-tuning process of LLMs. While we do not claim to improve general reasoning capabilities, RLSF leverages symbolic reasoning tools to improve downstream domain-specific tasks where syntax and semantics play a critical role. Our results show a significant improvement in all five tasks, over different traditional prompting and fine-tuning methods. Notably, the RLSF-tuned galactica-1.3b achieves superior results compared to GPT-4 (1000× larger) on the three chemistry tasks, RLSF-tuned code-gemma-2b outperforms GPT-3.5 (100× larger) on the program synthesis task. Similarly, RLSF-tuned llama2-7b-chat also outperforms GPT-3.5 (25× larger) on Game of 24. Additionally, unlike traditional neuro-symbolic RL approaches, RLSF does not require differentiable reasoning systems, making it more versatile and practical.

**Limitations and future work.** This study demonstrates the initial potential of integrating symbolic feedback into RL frameworks, with empirical improvements in specific domains such as program synthesis, chemistry and mathematical tasks. While we do not aim to enhance the overall reasoning capabilities of LLMs, our focus has been on developing a new fine-tuning paradigm that outperforms traditional methods within specific domains. Future research may extend this to explore theoretical guarantees, its impact across other reasoning tasks, and broader LLM reasoning capabilities. Lastly, our focus has been solely on fine-tuning, but we believe that combining RLSF with multi-step symbolic feedback during inference could further boost performance.

## REPRODUCIBILITY STATEMENT

We have made several efforts to ensure the reproducibility of our work. Section 3 provides a detailed explanation of the RLSF algorithm. Sections 4.1, 5.1, 6.1 cover the experimental setup, including the benchmarks used for RLSF in the five different tasks. In Section D, we outline the computational resources used and the implementation details, including the framework used to implement RLSF. In addition, the supplementary materials include an anonymous version of the source code as well as any modified datasets to facilitate reproducibility.

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

APPENDIX

# A ADDITIONAL DETAILS FOR REASONING TASK A: NATURAL LANGUAGE PSEUDO-CODE TO C++ CODE

## A.1 IMPLEMENTATION DETAILS

For reasoning task A of translating pseudo-code in natural language to C++ code, we used the SPoC dataset. However, the codes in the SPoC dataset lack `#include` preprocessor directives and `using namespace` lines. As such, during fine-tuning, the LLM is not trained to generate such lines. To resolve this, we prepend a reasonably comprehensive and fixed set of 21 preprocessor directives along with a `using namespace std;` line to each LLM-generated code snippet before calculating CompAcc, FCorrAcc, and the token-level feedback. This ensures that the LLM-generated code compiles correctly with the `g++` compiler.

## A.2 ADDITIONAL RESULTS

Most state-of-the-art code generation LLMs (Wang et al., 2022; Le et al., 2022; Shojaee et al., 2023; Dou et al., 2024; Wang et al., 2024a) employing RL-based fine-tuning rely on either Boolean compiler feedback (e.g., $+1$ if the code compiles, $-1$ if it does not), Boolean success feedback (e.g., $+x$ if all test cases pass, $-y$ if at least one test case fails), or a combination of both. In our natural language pseudo-code to code translation experiments, presented in Table 1, our aim was to compare these conventional binary scalar reward-based methods with the proposed fine-grained vector reward in RLSF.

For this comparison, we evaluated two types of scalar rewards: *success feedback* and *compiler feedback*, consistent with the binary reward schemes commonly used in state-of-the-art methods. Notably, success feedback provides an even sparser signal than compiler feedback. This is because the likelihood of LLM-generated code either (1) passing all test cases vs. failing at least one test case or (2) successfully compiling vs. failing to compile, results in a more balanced distribution in the second scenario. In other words, the disparity between success and failure cases is higher for success feedback compared to compiler feedback. As shown in Table 4, LLMs fine-tuned with Boolean success feedback underperform compared to those fine-tuned with Boolean compiler feedback.

The proposed vectorized feedback in RLSF enhances the binary scalar feedback schemes used in state-of-the-art methods in two key ways: (a) it refines the success reward by incorporating the number of test cases passed, moving beyond a binary outcome and, (b) it introduces a vectorized symbolic feedback mechanism that assigns lower rewards to tokens in code lines flagged by the compiler and higher rewards to tokens in unflagged lines. These innovations provide a more nuanced and effective fine-tuning paradigm for code generation LLMs, as demonstrated by our results.

Table 4: **Additional Results for Natural Language Pseudo-code to Code Translation:** Comparison of scalar feedback schemes and vectorized feedback in RLSF for fine-tuning LLMs.

| LLM | Configuration | CompAcc (%) | FCorrAcc (%) | Pass@1 (%) |
|---|---|---|---|---|
| `code-gemma-2b` (CodeGemma-Team, 2024) | SFT + RL (with Boolean success f/b) | 50.11 | 22.30 | 10.29 |
| | SFT + RL (with Boolean compiler f/b) | 54.89 | 24.71 | 12.77 |
| | SFT + *RLSF (with token-level f/b)* | **63.95** | **41.30** | **28.80** |
| `stable-code-instruct-3b` (Pinnaparaju et al., 2024) | SFT + RL (with Boolean success f/b) | 40.04 | 7.82 | 6.69 |
| | SFT + RL (with Boolean compiler f/b) | 48.43 | 10.91 | 9.22 |
| | SFT + *RLSF (with token-level f/b)* | **54.27** | **18.44** | **16.09** |
| `deepseek-coder-1.3b-base` (Guo et al., 2024) | SFT + RL (with Boolean success f/b) | 19.12 | 6.92 | 3.43 |
| | SFT + RL (with Boolean compiler f/b) | 19.97 | 8.07 | 3.88 |
| | SFT + *RLSF (with token-level f/b)* | **38.92** | **25.89** | **14.51** |

# B ADDITIONAL DETAILS FOR REASONING TASKS B: CHEMISTRY

## B.1 OVERVIEW OF THE THREE CHEMISTRY TASKS

**Forward synthesis** involves predicting the product of a chemical reaction based on given reactants and reagents. In computational chemistry, forward synthesis prediction allows for the simulation of

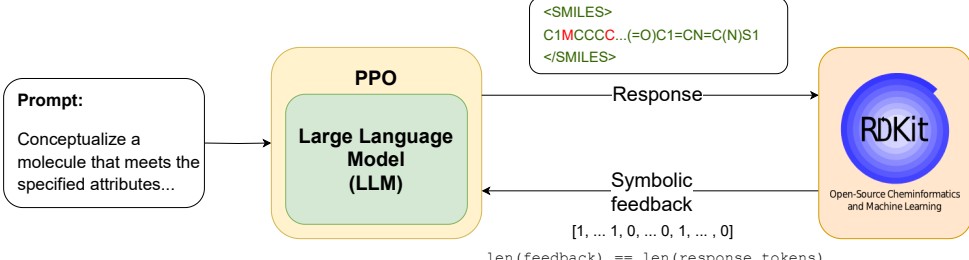

Figure 3: **RLSF for one of the chemistry tasks - Molecule Generation:** In this illustration, the symbolic environment utilizes RDKit (RDKit, 2023) to generate a token-level reward vector as feedback based on the presence or absence of any syntactical errors. Moreover, for the semantic errors, we again use RDKit to check for the presence of the required functional groups mentioned in the input natural language description and penalize the entire generated molecule if it lacks the required functional groups. Each element in the reward vector corresponds to a token in the response, where erroneous tokens are penalized with a value of 0 and correct ones are assigned 1. The last element of the reward vector (corresponding to the `<EOS>` token) is 1 only if the entire response is correct, otherwise, it is 0.

chemical reactions, allowing chemists to plan experiments without conducting them physically. For example, given the reactants phenoxazine (`NC1=CC=C2OCOC2=C1`) and formaldehyde (`O=CO`), the model predicts the product as `O=CNC1=CC=C2OCOC2=C1`.

**Retrosynthesis** involves determining the reactants required to create a specific product. It is essential for planning synthetic pathways, especially for complex molecules. For instance, the product `CC1=CC=C(N)N=C1N` can be derived from the reactants `CC(C#N)CCC#N` and ammonia (`NH3`).

**Molecular generation** involves generating a molecule that meets specific requested chemical and biological properties in natural language. This process is widely utilized in drug discovery and materials science to create compounds with specific characteristics, such as the presence of certain functional groups, binding affinity, stability, or bioactivity. For example, a molecule described as "a red-colored pigment with antibiotic properties..." is generated as `CCCCCC1=C(C)NC(/C=C2\N=C(C3=CC=CN3)C=C2OC)=C1`.

### B.2 DATASET SPLIT

We focus on three key tasks: Molecule Generation, Forward Synthesis, and Retrosynthesis, each of which presents the challenge of having multiple valid outputs for a given input. We take datapoints from existing datasets (Lowe, 2017; Edwards et al., 2022; Fang et al., 2023; Yu et al., 2024a) where multiple outputs are possible for a given input, as this scenario presents a significant challenge for LLMs. In such cases, the model must not only generate syntactically valid outputs but also grasp domain-specific rules and context to accurately capture task details. Simple supervised fine-tuning (SFT), which directly maps inputs to outputs, often struggles with this complexity, relying more on memorization. By handling tasks with multiple potential outputs for a single input, the model is required to go beyond basic input-output mapping, forcing it to learn complex relationships within the data. This setup is crucial for testing the model's ability to generalize.

To ensure proper evaluation, we identify sample input-output pairs across tasks that correspond to the same molecules or reactions, and group these samples consistently in either the training, evaluation, or test set. In addition, samples with the same input but different outputs are handled with care. For instance, in the RS task, a single product can be synthesized from multiple sets of reactants, so we ensure that all samples with identical inputs remain in the same split to avoid data leakage. For the Molecule Generation task, we used 3367 training samples, 354 validation samples, and 933 test samples. The Forward Synthesis task consists of 10207 training samples, 1452 validation samples, and 2908 test samples. The Retrosynthesis task has 77561 training samples, 2051 validation samples, and 4382 test samples.

# C    ADDITIONAL DETAILS FOR REASONING TASK C: GAME OF 24 USING TREE OF THOUGHTS (TOT)

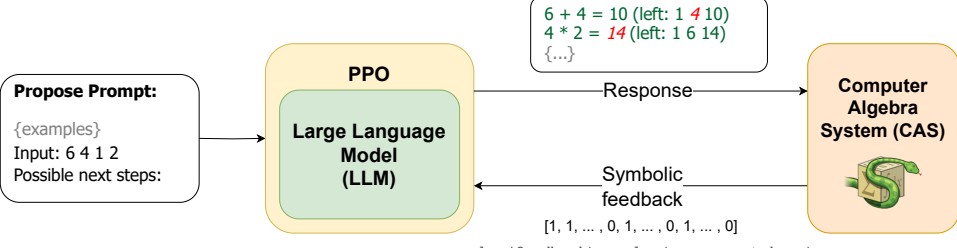

Figure 4: **RLSF for the Game of 24:** In this illustration, the symbolic environment utilizes the Computer Algebra System (CAS) library SymPy  (Meurer et al., 2017) to generate a token-level reward vector as feedback. Each element in the vector corresponds to a token in the response, where erroneous tokens are penalized with a value of 0 and correct ones are assigned 1. The last element of the reward vector (corresponding to the <EOS> token) is 1 only if the entire response is correct, otherwise, it is 0.

## C.1    GAME OF 24 USING TOT

To solve the Game of 24 using ToT, the process involves decomposing the problem into three steps, each representing an intermediate equation. Starting with the given input numbers, the LLM is prompted to propose possible next steps (or "thoughts") using a "propose prompt". Similar to Yao et al. (2024), we employ a breadth-first search (BFS) approach in ToT, maintaining the top 5 candidates at each step. Now, we prompt the LLM using the value prompt to evaluate the "thoughts." The score given by the LLM using the value prompt helps prune the "thoughts" generated by the "propose prompt." These two prompts are repeated, starting with the three remaining numbers (from all thoughts accepted after using the value prompt) to build the ToTs. This process is repeated until you arrive at the final equation that results in the number 24.

## C.2    CONVERTING THE SYMBOLIC FEEDBACK TO FINE-GRAINED SIGNAL

In Game of 24, we use a combination of SymPy and regex-based checks to validate model outputs and identify errors. SymPy performs syntax and semantic validation, ensuring the logical and mathematical correctness of the expressions, while regex is employed to verify numerical compliance with the game's rules.

Errors are classified into several categories for precise feedback. First, we detect instances where invalid numbers are used in the expression. For example, if the input numbers are (3, 8, 3, 3) and the model generates `3+8-3+10=18`, the use of 10 is flagged as an error because it is not part of the game input or an intermediate step.

Next, we verify the correctness of the equation using SymPy to ensure that the left-hand side equals the right-hand side of the equation. Discrepancies such as `3×(8+3)=27` are flagged as errors. Regex-based checks are also used to identify missing or extra numbers in the final expression. For example, if the input is [3, 8, 3, 3] and the model outputs `(3×8)+3=27`, one of the 3s is missing. In contrast, if the model produces `(3×8)+3+2`, the presence of extraneous 2 is flagged. Additional syntax issues, such as unbalanced parentheses, missing operators, or division by zero, are also detected and flagged.

To provide fine-grained feedback, we perform token-level pinpointing whenever possible, assigning penalties to erroneous tokens. For example, in the expression `(3×8)+3 3`, SymPy raises a syntax error due to the missing operator between the two 3s, and regex flags both 3s as repeated without justification. This allows errors to be traced back to specific tokens, enabling targeted corrections. This detailed feedback ensures that the model learns from precise and actionable signals, improving its ability to generate accurate solutions in subsequent attempts.

## D  IMPLEMENTATION, TRL LIBRARY, AND HARDWARE DETAILS

The RLSF algorithm is implemented by modifying the Transformer Reinforcement Learning (TRL) library (von Werra et al., 2020), a popular and comprehensive framework integrated with Huggingface transformers (Wolf et al., 2020) designed for training transformer language models with reinforcement learning (RL). We use the Proximal Policy Optimization (PPO) algorithm, commonly used in Reinforcement Learning from Human Feedback (RLHF) (Yang et al., 2024; Ouyang et al., 2022; Stiennon et al., 2020), for fine-tuning the language model. However, TRL only allows for scalar reward signals during the RL fine-tuning process. We modify the library to support reward vector (dense) signals, allowing fine-grained feedback at the token level. This enhancement enables RLSF to leverage symbolic feedback effectively during the RL process. We use LoRA (Hu et al., 2021) applied to all linear layers in the self-attention and FFN modules with `lora_r` and `lora_alpha` set to 16 during our fine-tuning. All our experiments were conducted on a high-performance CentOS V7 cluster equipped with Intel E5-2683 v4 Broadwell processors running at 2.10 GHz and 64 GiB of memory. We used 4 NVIDIA V100 GPUs for the tasks in Sections 4 and 5, and 1 NVIDIA V100 GPU for the task in Section 6.

