# OpenReview forum: "RLSF: Reinforcement Learning via Symbolic Feedback"
_ICLR.cc/2025/Conference — Submitted to ICLR 2025_

### Official Review · Reviewer_5FX7 · 2024-10-31

**Soundness:** 2
**Presentation:** 3
**Contribution:** 2
**Rating:** 6
**Confidence:** 2

**Summary:**

This paper presents a paradigm for finetuning large language models. In tasks where outputs are expected to belong to a certain formal language, symbolic programs can be used to check the validity / correctness and can be used as rewards to finetune an LLM. The authors demonstrate this approach on three problems - code generation, molecular generation / rethrosynthesis, and the so called game of 24.

**Strengths:**

1) The paper is written with good clarity. I am unsure about how original this work is because I am not aware of the literature in this area. But, I think the approach is fairly straightforward.
2) The results, especially in code generation tasks seem better than baselines like GPT-3.5.

**Weaknesses:**

1) Although there seems to be no glaring weakness to me, I am unsure about the significance of the chemistry problems that are used. I do not know how significant is Fingerprint Tanimoto Similarity or ~35%.
2) I am also not sure how novel this approach is. Can the authors point towards similar related work? Right now there is only one citation in the, "Neurosymbolic Reinforcement Learning (NRL)"  paragraph.

But both points are speculations rather than weaknesses, I would just appreciate an answer.

**Questions:**

Please look at the weakness section for questions.

---

> ### Author Response · Authors · 2024-11-18
>
> Thank you for your valuable feedback and questions. We appreciate your recognition of the clarity of our writing and the improved results in different tasks compared to baselines like GPT3.5 and GPT4. Below, we address your concerns regarding the novelty of our approach and the significance of the chemistry tasks.
>
> **Q1. Novelty and related work**
>
> **A1.** The Reinforcement Learning via Symbolic Feedback (RLSF) paradigm presents several innovative contributions that set it apart from existing methods. Section 2 of our paper talks about these points in more detail. We believe these references might help highlight the novelty of our work.
>
> 1. Fine-grained token-level symbolic feedback in LLMs: Unlike existing approaches that rely on scalar or binary signals [3,4,5], RLSF provides fine-grained, token-level feedback derived from poly-sized certificates generated by symbolic tools. This enables more precise error identification and correction, leading to significantly improved fine-tuning performance.
>
> 2. No need for differentiable symbolic feedback: Neurosymbolic Reinforcement Learning (NRL) approaches [1,2] typically require differentiable reasoning systems, limiting their flexibility. RLSF, by contrast, integrates non-differentiable symbolic tools into the environment, broadening its applicability while providing modular and interpretable feedback. The RLSF approach allows any LLM to be used as the agent and any symbolic tool as the environment, enabling fine-tuning of the LLM for specific tasks without requiring modifications to its architecture - unlike NRL, which necessitates architectural changes to accommodate symbolic tools.
>
> 3. Use in fine-tuning rather than in-context: The RLSF framework incorporates symbolic reasoning systems during fine-tuning, providing corrective feedback to the LLM. This approach contrasts with prior research [12,13], which only use symbolic tools during inference. By using reasoning systems during fine-tuning, RLSF enhances learning efficacy, enabling more granular adjustments to the model for the particular task.
>
> 4. Sound corrective feedback via symbolic systems: The reliance on symbolic reasoning tools ensures sound feedback, avoiding the pitfalls of self-verification or in-context feedback loops used by other LLM-based systems [6,7,8,9]. Unlike these approaches, RLSF addresses the limitations of LLMs in self-critiquing, as observed by some recent work [10,11], by relying on external symbolic verifiers (e.g., compilers, CAS systems), ensuring more accurate guidance.
>
> 5. Smaller and efficient models: By using symbolic feedback, RLSF significantly improves the performance of smaller models compared to closed-source, larger-scale models like GPT-3.5 and GPT-4 across five different tasks. This efficiency advantage is crucial for tasks requiring domain-specific understanding.

---

> ### Author Response · Authors · 2024-11-18
>
> **Q2. Significance of the chemistry tasks**
>
> **A2.** Tasks like molecular generation, forward synthesis, and retrosynthesis are foundational in drug discovery, materials science, and synthetic chemistry. Forward synthesis helps predict reaction outcomes, saving time and resources in planning experiments. Retrosynthesis is crucial for designing synthetic routes for complex molecules, including drugs. Molecular generation aids in creating novel molecules with desired properties, accelerating innovation in pharmaceuticals and materials science. These tasks are benchmarks for evaluating the ability of machine learning models to understand and navigate domain-specific constraints like chemical valence rules and reaction feasibility. The challenges of balancing structural accuracy, chemical validity, and task-specific performance make them ideal for testing the efficacy of fine-tuning techniques like RLSF.
>
> We appreciate the reviewer’s question regarding the significance of Fingerprint Tanimoto Similarity (FTS). The FTS score represents the structural similarity between the generated or predicted molecules and the reference molecules. In cheminformatics, FTS is a widely accepted metric for assessing molecular similarity [14,15].
>
> A score of 35% indicates moderate similarity, which is meaningful in tasks like molecular generation, forward synthesis and retrosynthesis. Such a score reflects the presence of shared structural features, such as functional groups or substructures, which are critical for determining chemical activity. In drug discovery and materials science, molecules with high similarity to known active compounds are often strong candidates for further exploration. Moreover, when combined with metrics like exact match (EM) and chemical validity, the FTS score highlights the ability of our method to generate valid and structurally meaningful molecules.
>
> It is important to note that the application of smaller language models to chemistry tasks is still in its infancy, with most prior work focusing on larger models. Smaller models have only recently started being applied to these complex tasks, and there is still a long way to go in advancing their capabilities. Our work represents an important stepping stone in this direction, demonstrating that smaller, fine-tuned models can produce meaningful outputs and compete with much larger counterparts. For example, we are able to outperform GPT4 using 1000x smaller models. This definitely paves the way for more accessible, efficient, and domain-specific AI applications in chemistry and related fields.
>
> We hope this clarifies the significance of FTS and how it is used to evaluate the quality of the chemistry tasks in our work.
>
> We also hope that our clarifications demonstrate the originality and significance of our work and respectfully request you to consider increasing your score if the clarifications address your concerns. Thank you again for your insights and questions.

---

> ### Author Response · Authors · 2024-11-18
>
> **References cited in the above rebuttal:**
>
> [1] Pascal Hitzler, Md Kamruzzaman Sarker, and Aaron Eberhart. Compendium of Neurosymbolic Artificial Intelligence, volume 369. IOS Press, 2023.
>
> [2] Kamal Acharya, Waleed Raza, Carlos Dourado, Alvaro Velasquez, and Houbing Herbert Song. Neurosymbolic reinforcement learning and planning: A survey. IEEE Transactions on Artificial Intelligence, 2023.
>
> [3] Yanju Chen, Chenglong Wang, Osbert Bastani, Isil Dillig, and Yu Feng. Program synthesis using deduction-guided reinforcement learning. In International Conference on Computer Aided Verification, pp. 587–610. Springer, 2020.
>
> [4] Jia Zou, Xiaokai Zhang, Yiming He, Na Zhu, and Tuo Leng. Fgeo-drl: Deductive reasoning for geometric problems through deep reinforcement learning. Symmetry, 16(4):437, 2024.
>
> [5] Prithwish Jana, Piyush Jha, Haoyang Ju, Gautham Kishore, Aryan Mahajan, and Vijay Ganesh. Cotran: An llm-based code translator using reinforcement learning with feedback from compiler and symbolic execution. arXiv preprint arXiv:2306.06755, 2023.
>
> [6] Noah Shinn, Federico Cassano, Ashwin Gopinath, Karthik Narasimhan, and Shunyu Yao. Reflexion: Language agents with verbal reinforcement learning. Advances in Neural Information Processing Systems, 36, 2024.
>
> [7] Aman Madaan, Niket Tandon, Prakhar Gupta, Skyler Hallinan, Luyu Gao, Sarah Wiegreffe, Uri Alon, Nouha Dziri, Shrimai Prabhumoye, Yiming Yang, et al. Self-refine: Iterative refinement with self-feedback. Advances in Neural Information Processing Systems, 36, 2024.
>
> [8] Debjit Paul, Mete Ismayilzada, Maxime Peyrard, Beatriz Borges, Antoine Bosselut, Robert West, and Boi Faltings. Refiner: Reasoning feedback on intermediate representations. arXiv preprint arXiv:2304.01904, 2023.
>
> [9]Geunwoo Kim, Pierre Baldi, and Stephen McAleer. Language models can solve computer tasks. Advances in Neural Information Processing Systems, 36, 2024.
>
> [10] Yujun Mao, Yoon Kim, and Yilun Zhou. Champ: A competition-level dataset for fine-grained analyses of llms’ mathematical reasoning capabilities. arXiv preprint arXiv:2401.06961, 2024.
>
> [11] Kaya Stechly, Karthik Valmeekam, and Subbarao Kambhampati. On the self-verification limitations of large language models on reasoning and planning tasks. arXiv preprint arXiv:2402.08115, 2024.
>
> [12] Lakshya A Agrawal, Aditya Kanade, Navin Goyal, Shuvendu Lahiri, and Sriram Rajamani. Monitor-guided decoding of code lms with static analysis of repository context. Advances in Neural Information Processing Systems, 36, 2024.
>
> [13] Subbarao Kambhampati, Karthik Valmeekam, Lin Guan, Kaya Stechly, Mudit Verma, Siddhant Bhambri, Lucas Saldyt, and Anil Murthy. Llms can’t plan, but can help planning in llm-modulo frameworks. arXiv preprint arXiv:2402.01817, 2024.
>
> [14] Botao Yu, Frazier N Baker, Ziqi Chen, Xia Ning, and Huan Sun. Llasmol: Advancing large language models for chemistry with a large-scale, comprehensive, high-quality instruction tuning dataset. arXiv preprint arXiv:2402.09391, 2024a.
>
> [15] Cao, H., Liu, Z., Lu, X., Yao, Y., & Li, Y. Instructmol: Multi-modal integration for building a versatile and reliable molecular assistant in drug discovery. arXiv preprint arXiv:2311.16208, 2023.

---

> > ### Comment · Reviewer_5FX7 · 2024-11-19
> >
> > Thank you for addressing my questions. I think the current rating is fair given that the approach seems to be an incremental advancement. The idea that fine-grained, token-level feedback would be better than a scalar delayed feedback very well known. The contribution of the paper is to validate this well known idea.

---

> > > ### Author Response · Authors · 2024-11-22
> > >
> > > Thank you for your feedback. We respectfully disagree with the characterization of our work as merely validating a well-known idea. We want to emphasize that our key contribution lies in "leveraging symbolic feedback" (i.e., interpretable, token-level corrections via poly-sized certificates from symbolic reasoning tools) to obtain fine-grained feedback. This is the first demonstration of integrating symbolic tools in RL post-training to produce such detailed and domain-specific rewards. The novelty is not in the general notion of fine-grained feedback but in creating a framework to systematically extract and utilize symbolic feedback in tasks like program synthesis, chemistry, and arithmetic tasks, which has not been shown before. We hope this distinction clarifies the originality and significance of our work.

---

> > > > ### Comment · Reviewer_5FX7 · 2024-11-22
> > > >
> > > > I apologize for the characterizing the main contribution as just the "validation of one well known result". I do agree there are many interesting experiments and insights that one could draw from the paper.
> > > >
> > > > I do agree that the precise way of "leveraging symbolic feedback" is a contribution, it can still be considered as an incremental advancement, that comes under the very broad umbrella of papers that demonstrate the importance of dense feedback. Because of this, I think the current rating (marginally above the acceptance) is a fair.

---

> > > > > ### Author Response · Authors · 2024-11-24
> > > > >
> > > > > Thank you for your thoughtful response and for acknowledging the broader insights and contributions of our work. We appreciate your recognition of the novelty of leveraging symbolic feedback. While we understand your perspective on its incremental nature, our method's unique integration of symbolic reasoning tools provides a meaningful step forward, especially in its applicability to diverse domains.
> > > > >
> > > > > We respect your evaluation and appreciate your feedback in helping us improve our manuscript. Thank you for your time and consideration.

---

### Official Review · Reviewer_sFgh · 2024-11-04

**Soundness:** 3
**Presentation:** 2
**Contribution:** 2
**Rating:** 6
**Confidence:** 3

**Summary:**

This paper proposes RLSF or RL with symbolic feedback. RLSF's main difference with RLHF are two components. 1) A symbolic feedback generator that produces poly-sized certificates which basically means the feedback is not too unreasonably large compared to the input, .i.e is polynomial in size of the input. An examples of these symbolic feedback generator is the output of a compiler when compiling a program. 2) A function that takes in this feedback, the policy-sized certificate, and translates it to per-token rewards, e.g. assigning negative rewards to the lines of code that had syntax problems. Learning from this symbolic feedback improves RL post-training of LLMs on various tasks.

**Strengths:**

It is not clear how to integrate symbolic feedbacks into a RL post-trainng of LLMs. Therefore, it seems RL post-training is missing out those feedbacks and finding a way to integrate them into the training loop should have advantages. This work is a first attempt at doing these which is valuable by trying to translate this feedbacks to environment rewards. It is different from other works that want to give the LLM a way to use symbolic tools.  Also, the work tests the idea on diverse tasks: Translation of Pseudo-Code to Code, Modelling chemical reactions, and game of 24 as a toy example. The work shows token-level rewards given by this system improves performance especially in terms of lexical or sytax errors.

**Weaknesses:**

In terms of novelty, the fact that fine-grained feedback works better is well-known in RL community. For examples, There is a plethora of works on reward shaping. However, I know that in NLP, people know RLHF with its boolean feedback. Also in RL for reasoning we have papers that try to incorporate process based rewards instead of scalar rewards. Therefore, showing fine-grained signal works better is not considered very novel. However, I have not seen another work that tries to somehow extract and translate token-level feedback from these symbolic feedbacks. That is the the novelty of the paper. I think this paper has an interesting message for people who are doing RL post-training to try to incorporate rewards of previously symbolic systems which I think is important and is a strength.

In terms of the experimentation, I think there is a big problem in the coding experiments. It may be that I misunderstood but I want to double check. After the SFT is done on the models, it seems the actual comparison is between training with the binary feedback, and the then the symbolic feedback. The main problem is that the symbolic feedback also contains the actual rewrad of whether the test cases are passed or not while the binary feedback is just whether the program compiles or not. That is not the correct comparison as the second option has the access to the actual success reward on the tasks in top of the integration of the symbolic feedbacks. The comparison that I think shows the significance of the inclusion of the symbolic feedback is to compare A) training with the success reward which can include the compilability reward as well  B) training with the success reward augmented with symbolic rewards.

I don’t have an expertise in the tasks involving chemical reactions. I have asked a few questions about them to make sure the signficant of the contribution.

In terms of writing, I think the paper is not written in terms with its actual message. I think the paper is written from the angle that its message is that “fine-grained signal is better”. However, this is a well-known message and not surprising. The actual hard task is how to transform a symbolic feedback to a fine-grained signal. If the paper was written about the details and challenges of this conversion I think its contribution would be much more apparent. I think the authors have actually done the hard work of the details and nitty-gritty of this conversion but decided to put the focus on the RLSF rather than on those details. There are many questions about how-to of this extraction of signal from symbolic feedback to fine-grained signal that I think it deserves a well written paper to discuss them. I have asked them in the questions section and I would like a lot if the authors include a section about that in their paper.

**Questions:**

1-In the pseudo-code to code translation experiments, is there a reason that you did not have a run with the success reward (maybe alongside compilability reward)? What happens in that experiments and is there still benefit to integrate the symbolic feedback from the compiler?

2-What are the challenges and details of the choices for converting the symbolic feedback to per-token level rewards? I understand when a line does not compile, we can put negative rewards on that line. But is it always correct? If I am not wrong, there are cases in which a line does not compile because of a line before is not written. For example, a variable is not defined or a space or newline is forgotten. Isn’t it possible that with this the conversion actually misses the actual source of the problem and keep it untouched? I think there is some degree of credit assignment going on and the assignment is based on best guesses. While I am not familiar with the chemistry tasks, I am wondering what the output of the symbolic feedback generators look like and how easy it is to transform to per-token rewards? Also, I did not follow the explanation of the paper on how they integrated the algebraic system to the game of 24. Can you explain this in detail? I think the details of how to cover these signal is the actual contribution of the paper.

3-For modelling the chemical reactions, you take out a pre-trained language model and then fine-tune it. Is that correct? I am just wondering if this is standard practice because these reactions are very far from written human texts and I was wondering if the standard practice is to just train from scratch? Can you cite other papers that also employ the same strategy?

---

> ### Author Response · Authors · 2024-11-18
>
> We thank the reviewer for their thoughtful and detailed feedback, as well as for recognizing the novelty and importance of integrating symbolic feedback into RL post-training. We appreciate your recognition of the strengths of our work, including the innovative approach, diverse task evaluations, and significant improvements. Below, we address your questions, provide clarifications to further highlight the contributions of our work, and show results from additional experiments that we performed.
>
> **Q1. Separate section on how to transform a symbolic feedback to a fine-grained signal: ...paper is not written in terms with its actual message...The actual hard task is how to transform symbolic feedback to a fine-grained signal.**
>
> **A1.** We sincerely thank the reviewer for this excellent suggestion! You are absolutely correct that one of our key contributions lies in the mechanism for extracting symbolic feedback and converting it into per-token rewards. As noted by the reviewer, there is no prior work on using symbolic systems to provide such granular feedback in RL fine-tuning of LLMs. This is a significant contribution to RL post-training, as symbolic feedback is more interpretable and domain-specific than traditional scalar rewards.
>
> In Section 3 of the paper, we discussed how symbolic feedback can be transformed into a fine-grained signal. At the end of Section 3, we also addressed the generalizability of RLSF, which stems from its modular design, ensuring that it can be applied to any task where outputs are expressible in a formal language and symbolic tools are available to offer segment-wise feedback. In response to the reviewer’s suggestions, we will include a dedicated section in the final version to offer a deeper discussion on this transformation process and better position the paper to emphasize this key contribution.

---

> ### Author Response · Authors · 2024-11-18
>
> **Q2. In the pseudo-code to code translation experiments, is there a reason that you did not have a run with the success reward (maybe alongside compilability reward)? What happens in that experiments and is there still benefit to integrate the symbolic feedback from the compiler?**
>
> **A2.** The reviewer is absolutely correct in pointing out that, in our experiments, we did not use the success reward for binary feedback but instead relied on the compilation reward. Most state-of-the-art code generation LLMs [1,2,3,4,5] employing RL-based fine-tuning use either Boolean compiler feedback (+1 if the code compiles, -1 if it doesn’t), Boolean success feedback (+x if all test cases pass, -y if at least one fail), or a combination of these two. In our pseudo-code-to-code translation experiments, our goal was to compare these conventional binary scalar reward-based methods with the proposed fine-grained vector reward of RLSF.
>
> Second, when comparing scalar rewards versus our proposed vectorized reward, we considered two types of scalar rewards: success reward and compilability reward, consistent with the binary feedback employed in state-of-the-art methods. It is worth noting that the success reward is an even sparser signal compared to the compilability reward. This is because the likelihood of LLM-generated code either (1) passing all test cases vs. failing at least one test case or (2) successfully compiling vs. failing to compile, results in a more balanced distribution in the second case. The disparity between success and failure classes is higher for the first case than the second. Therefore, we opted for the binary compilability reward as the scalar feedback for our comparisons.
>
> However, to address the reviewer's concern, we conducted experiments using the success reward as the scalar feedback and obtained the following results.
>
> code-gemma-2b (SFT + RL (with binary compilation f/b)): CompAcc = 54.89%, FCorrAcc = 24.71%, Pass@1 = 12.77%
>
> code-gemma-2b (SFT + RL (with binary success f/b)): CompAcc = 50.11%, FCorrAcc = 22.30%, Pass@1 = 10.29%
>
> stable-code-instruct-3b (SFT + RL (with binary compilation f/b)): CompAcc = 48.43%, FCorrAcc = 10.91%, Pass@1 = 9.22%
>
> stable-code-instruct-3b (SFT + RL (with binary success f/b)): CompAcc = 40.04%, FCorrAcc = 7.82%, Pass@1 = 6.69%
>
> deepseek-coder-1.3b-base (SFT + RL (with binary compilation f/b)): CompAcc = 19.97%, FCorrAcc = 8.07%, Pass@1 = 3.88%
>
> deepseek-coder-1.3b-base (SFT + RL (with binary success f/b)): CompAcc = 19.12%, FCorrAcc = 6.92%, Pass@1 = 3.43%
>
> As shown, the performance of LLMs fine-tuned with binary success reward is worse than that fine-tuned with the binary compilability reward. We would be glad to include this new result in Table 1 and incorporate the aforementioned discussion into the paper.
>
> Our approach enhances the binary scalar rewards used in state-of-the-art methods in two key ways: (a) We make the success reward more fine-grained by incorporating the number of test cases passed, rather than relying on a binary outcome. (b) We propose a vectorized symbolic feedback mechanism where we assign lower rewards to tokens of code lines flagged by the compiler and higher rewards to tokens in other lines. These innovations enable a more nuanced and effective fine-tuning paradigm for code generation models, as demonstrated in our results.

---

> ### Author Response · Authors · 2024-11-18
>
> **Q3. ...when a line does not compile, we can put negative rewards on that line. But is it always correct?**
>
> **A3.** Modern compilers are generally proficient at producing accurate error messages that closely correspond to the actual source of the problem. In our method, we treat the lines flagged by the compiler as indicators of where the syntactical issues are likely to reside in the generated code. We empirically observed that in LLM-generated codes, these flagged lines typically provide the most accurate starting point for identifying the error, as they represent the earliest point at which the compiler detects a problem. Consequently, in most cases involving LLM-generated code, the problem can often be resolved by modifying the flagged line itself. This is why assigning negative rewards to the flagged lines proves beneficial. We have demonstrated through our experiments that this approach is both practical and reliable for assigning fine-grained feedback, effectively guiding the model to iteratively improve code quality.
>
> It is possible for a line to be flagged as erroneous by the compiler due to an issue in a preceding line. Having said that, our empirical observations suggest that such occurrences are infrequent in LLM-generated code.

---

> ### Author Response · Authors · 2024-11-18
>
> **Q4. ...what the output of the symbolic feedback generators look like and how easy it is to transform to per-token rewards? ...explanation of the paper on how they integrated the algebraic system to the game of 24. Can you explain this in detail? I think the details of how to cover these signal is the actual contribution of the paper.**
>
> **A4.** In our chemical modeling tasks, the symbolic feedback generator is RDKit, a widely used cheminformatics toolkit. RDKit analyzes the model’s output (e.g., SMILES strings) and generates error logs, which are transformed into per-token rewards. Let's look at one such example. Let's say the model generates the SMILES string C(C)(C)N(C)(C)(C), RDKit identifies an issue with the nitrogen atom, which violates valence rules. The symbolic feedback generator (RDKit) attempts to parse the input SMILES string and we get an error saying, "Explicit valence for atom # 3 N, 4, is greater than permitted." This error indicates that the nitrogen atom (N, atom index 3 in the molecule) has a valence of 4, exceeding the maximum allowed for nitrogen in this context. In order to correct the SMILES string, one can either add a positive charge to nitrogen if the molecule is intended to be a quaternary ammonium ion, i.e., Chem.MolFromSmiles('C(C)(C)[N+](C)(C)(C)') or reduce the number of bonds to nitrogen to make it chemically valid, e.g., trim one methyl group: Chem.MolFromSmiles('C(C)(C)N(C)(C)'). So, the erroneous tokens are N (penalized for being the central atom exceeding valence limits) and parenthetical C groups after N (penalized as contributors to the invalid valence state.).
>
> Beyond valence errors, RDKit detects a wide range of issues (more details in https://www.rdkit.org/docs/Cookbook.html), including incorrect atoms (hallucinations), syntax errors (e.g., invalid characters, unclosed rings, unbalanced parentheses), and semantic violations (e.g., the first law of chemistry, missing functional groups in molecule generation). RDKit's error messages follow a consistent format, allowing us to parse the natural language descriptions and categorize errors for targeted processing.
>
> An example of a semantic check would be for forward synthesis tasks, we ensure adherence to the first law of chemistry by penalizing outputs that introduce new atoms not present in the reactants. For example, if the reactants contain hydrogen and carbon atoms, but the product contains an unexpected nitrogen atom, this would trigger a penalty. Similarly, for molecule generation, we verify the presence of specified functional groups and penalize outputs that lack them. This token-level feedback allows the model to learn corrections in a targeted manner, addressing both syntax and semantic errors effectively. By incorporating these detailed checks, our approach ensures that the symbolic feedback aligns closely with domain-specific requirements.
>
> In the Game of 24, we combine SymPy with regex-based checks to validate model outputs and identify errors. SymPy handles syntax and semantic validation, while regex ensures numerical correctness based on the game’s rules. Errors include: (1) Invalid number {number} used in the expression: This occurs when the expression includes numbers not part of the game inputs or carried over from intermediate steps (e.g., input numbers are [3, 8, 3, 3], and the model generates 3 + 8 + 10 -- 10 triggers an error). (2) Expression {left_hand_side} does not equal {right_hand_side}: SymPy validates that the equation equals 24, flagging discrepancies like 3 * (8 + 3) = 27. (3) Numbers {missing_numbers} are missing in the last line: Regex identifies cases where numbers from the input are not used (e.g., input [3, 8, 3, 3], model generates (3 * 8) + 3, omitting one 3). (4) Numbers {extra_numbers} are extra in the last line: This occurs when extraneous numbers are introduced in the final solution (e.g., (3 * 8) + 3 + 2). (5) Numbers {incorrect_numbers} are incorrect in the last line, expected {expected_numbers}: This happens when valid numbers are replaced by hallucinated values (e.g., input [3, 8, 3, 3], output (3 + 8 + 7), where 7 replaces a valid 3). (6) Other syntax checks: Unbalanced parenthesis, missing operator, division by zero errors. We perform token-level pinpointing (when applicable), to penalize erroneous tokens. For instance, in (3 * 8) + 3 3, SymPy raises a syntax error due to the missing operator between 3 and 3, and regex flags both the 3's as repeated without justification. Error is traced back to specific tokens for targeted corrections. This detailed feedback ensures the model learns precise corrections.

---

> ### Author Response · Authors · 2024-11-18
>
> **Q5. For modelling the chemical reactions, you take out a pre-trained language model and then fine-tune it. Is that correct? I am just wondering if this is standard practice because these reactions are very far from written human texts and I was wondering if the standard practice is to just train from scratch? Can you cite other papers that also employ the same strategy?**
>
> **A5.** Yes, fine-tuning pre-trained LLMs for chemical reaction modeling is a recognized and effective approach in computational chemistry. This strategy uses the general language understanding of LLMs and adapts them to the specific syntax and semantics of chemical data, such as SMILES representations during fine-tuning. LlaSMol [6] and Mol-Instructions [7] are two of the most recent papers that use this approach.
>
> Thank you very much for your valuable feedback. We greatly appreciate your insights and would be delighted to incorporate these clarifications and changes into the paper to further enhance its clarity. Additionally, we kindly request you to consider increasing your score if the clarifications address your concerns.
>
> **References:**
>
> [1] Wang, X., Wang, Y., Wan, Y., Mi, F., Li, Y., Zhou, P., ... & Liu, Q. (2022). Compilable neural code generation with compiler feedback. arXiv preprint arXiv:2203.05132.
>
> [2] Le, H., Wang, Y., Gotmare, A. D., Savarese, S., & Hoi, S. C. H. (2022). Coderl: Mastering code generation through pretrained models and deep reinforcement learning. Advances in Neural Information Processing Systems, 35, 21314-21328.
>
> [3] Shojaee, P., Jain, A., Tipirneni, S., & Reddy, C. K. (2023). Execution-based code generation using deep reinforcement learning. arXiv preprint arXiv:2301.13816.
>
> [4] Dou, S., Liu, Y., Jia, H., Xiong, L., Zhou, E., Shen, W., ... & Gui, T. (2024). Stepcoder: Improve code generation with reinforcement learning from compiler feedback. arXiv preprint arXiv:2402.01391.
>
> [5] Wang, Y., Wang, Y., Guo, D., Chen, J., Zhang, R., Ma, Y., & Zheng, Z. (2024). Rlcoder: Reinforcement learning for repository-level code completion. arXiv preprint arXiv:2407.19487.
>
> [6] Botao Yu, Frazier N Baker, Ziqi Chen, Xia Ning, and Huan Sun. Llasmol: Advancing large language models for chemistry with a large-scale, comprehensive, high-quality instruction tuning dataset. COLM 2024.
>
> [7] Fang, Yin, Xiaozhuan Liang, Ningyu Zhang, Kangwei Liu, Rui Huang, Zhuo Chen, Xiaohui Fan, and Huajun Chen. "Mol-instructions: A large-scale biomolecular instruction dataset for large language models." ICLR 2024.

---

> > ### Author Response · Authors · 2024-11-22
> >
> > Dear Reviewer,
> >
> > If there are any remaining questions or points of clarification needed after our rebuttal, we would be happy to address them. We kindly remind you to share any further thoughts or responses so that we can ensure all concerns are fully addressed. Thank you for your time and consideration.

---

> > > ### Comment · Reviewer_sFgh · 2024-11-23
> > >
> > > I thank the authors for their feedback.
> > >
> > > However, I should note that the feedback was very long. I read through it. I really liked the explanations about the exact way the signal is translated to a fine-grained one, especially the chemical task.  I will increase my score but only if the authors add this to the main paper. These are absolutely useful and very important.

---

> > > > ### Author Response · Authors · 2024-11-24
> > > >
> > > > Thank you for your thoughtful feedback and for taking the time to read through our response. We greatly appreciate that you found the explanations about translating symbolic feedback into fine-grained signals, particularly for the chemical tasks, both useful and important.
> > > >
> > > > We completely agree that these details needed to be included in the main paper. It was an oversight on our part not to include them initially, and we have since made the necessary changes in the updated version of the manuscript, incorporating the explanations and examples to ensure clarity and completeness. Particularly, we have updated Sections 5.1 and 6.1 and added Appendix C.2 to provide clearer explanations, along with detailed examples, of how symbolic feedback is converted into fine-grained signals for the chemistry tasks and the Game of 24.
> > > >
> > > > Additionally, we have also updated Section 4.4 and added Appendix A.2 to include the experiments suggested by you comparing the use of binary success reward and binary compilability reward in pseudo-code to code translation task.
> > > >
> > > > We also sincerely thank you for considering increasing your score and for your constructive feedback, which has definitely helped improve the quality of our paper.

---

> > > > > ### Comment · Reviewer_sFgh · 2024-11-25
> > > > >
> > > > > Just for clarification, I already increased my score from 5->6 having confidence that you will update the paper! Thanks.

---

> > > > > > ### Author Response · Authors · 2024-11-25
> > > > > >
> > > > > > Thank you!

---

### Official Review · Reviewer_T2Ku · 2024-11-04

**Soundness:** 3
**Presentation:** 3
**Contribution:** 3
**Rating:** 6
**Confidence:** 2

**Summary:**

This paper propose a new fine-tuning paradigm for LLM post-training called RLSF, where the reasoning tools can provide feedback to the LLMs via poly-sized certificates characterizing errors in the LLM-generated object with respect to some correctness specification.
RLSF is evaluated across five different tasks, demonstrating superior performance compared to traditional fine-tuning methods.

**Strengths:**

* Innovativeness: RLSF is the first reinforcement learning paradigm that utilizes multi-dimensional certificates generated by symbolic reasoning tools to provide fine-grained feedback, addressing the limitations of traditional reward models.

* Efficiency: RLSF significantly enhances the performance of LLMs across multiple tasks, particularly excelling in the conversion of natural language pseudocode to C++ and in chemical tasks.

* Practicality: RLSF does not require a differentiable symbolic reasoning system, which increases its flexibility and applicability in real-world scenarios.

* Experimental Validation: The article validates the effectiveness and superiority of RLSF through experiments on five different tasks, showcasing its potential in specific domain tasks.

**Weaknesses:**

* The proposed RLSF relies on symbolic reasoning tools to generate feedback, and the performance and availability of these tools may influence the effectiveness of RLSF. While the article mentions that these tools perform well in practice, it does not provide a detailed discussion of their limitations and possible alternatives.
* The abstract section is overly lengthy and could benefit from a more concise phrasing.

**Questions:**

* The RLSF method demonstrates exceptional performance on specific tasks, However, its applicability to a broader range of reasoning tasks or general tasks remains unclear. Is there theoretical or empirical evidence supporting its effectiveness across different tasks?
* Does the performance of symbolic tools decline when handling complex or large-scale problems? Are there alternative solutions or improvements that can be considered?

---

> ### Author Response · Authors · 2024-11-18
>
> Thank you for your thoughtful and constructive feedback on our work. We found your review highly insightful and valuable. We greatly appreciate your recognition of the strengths of our proposed RLSF framework, particularly its innovativeness, practical applicability, and strong empirical validation. Below, we address your concerns and questions regarding broader applicability, symbolic tool performance, and scalability.
>
> **Q1. Applicability to broader reasoning tasks and theoretical analysis**
>
> **A1.** Our results provide strong empirical evidence of RLSF’s adaptability to tasks with formal outputs. The modular nature of RLSF ensures its applicability to any task with outputs expressible in a formal language, and availability of symbolic tools to validate those outputs. We discuss this in more detail at the end of Section 3 in the paper. In fact, we have already started exploring RLSF-like techniques for proof synthesis, symbolic regression, material science, code repair, and physics simulators that could extend their applicability to deeper reasoning tasks.
>
> The three distinct domains - code generation, chemistry, and mathematical reasoning (Game of 24) - were chosen for their diverse requirements. Code generation is a well-studied task in symbolic reasoning, showing how RLSF improves both compilation and functional correctness metrics. Chemistry is a challenging domain where outputs must adhere to strict syntax and semantic rules, demonstrating the versatility of RLSF. Mathematical Game of 24 reasoning task highlights the benefits of symbolic feedback in guiding LLM-generated arithmetic solutions. We selected these domains because they are benchmarks for tasks with formal outputs requiring adherence to domain-specific constraints. By demonstrating improvements across these diverse domains, we illustrate RLSF’s generalizability and robustness. Additionally, tasks like chemistry and mathematics are relatively underexplored with smaller models, and our results contribute to advancing their applicability.
>
> We discuss the lack of deeper theoretical guarantees in the Limitations section, noting that this is an area for future work. As RLSF is the first paradigm to integrate token-level feedback with symbolic tools in this manner, we consider our work an important stepping stone, providing empirical evidence to guide subsequent theoretical advancements. We have demonstrated empirical performance across domains and consistent improvements over larger models such as GPT-4 highlight the substantial contribution of RLSF.
>
> **Q2. Performance of symbolic reasoning tools in complex or large-scale scenarios**
>
> **A2.** Fortunately, most of the verification tools utilized in our framework are widely used systems optimized for real-world applications built to handle large and complex inputs with high efficiency. Examples include compilers for code verification, cheminformatics libraries like RDKit for molecular analysis, and symbolic math tools for reasoning tasks, all of which are capable of processing intricate and high-dimensional data. Empirically, we observed that these tools scaled effectively across all tasks evaluated in our study, even in scenarios involving complex inputs such as large codebases, diverse chemical structures, and multi-step reasoning problems. This empirical evidence demonstrates their robustness and ability to integrate seamlessly with the RLSF framework, providing accurate and timely feedback without introducing significant computational overhead.
>
> **Q3. Abstract too long**
>
> **A3.** We also appreciate your feedback on the abstract. We will revise it to be more concise, focusing on the essential contributions and findings.
>
> We hope our responses address your concerns and provide clarity about the generalizability and scalability of RLSF. As you have noted, RLSF introduces a novel paradigm with token-level feedback, modular design, and demonstrated effectiveness across multiple domains. These strengths, coupled with the flexibility and potential for future expansion, highlight RLSF’s contribution as a significant advancement in fine-tuning LLMs for domain-specific tasks.
>
> Thank you once again for your thoughtful feedback, and we respectfully request you to consider increasing your score if the clarifications address your concerns.

---

> > ### Author Response · Authors · 2024-11-22
> >
> > Dear Reviewer,
> >
> > If there are any remaining questions or points of clarification needed after our rebuttal, we would be happy to address them. We kindly remind you to share any further thoughts or responses so that we can ensure all concerns are fully addressed. Thank you for your time and consideration.

---

### Official Review · Reviewer_bSi8 · 2024-11-10

**Soundness:** 2
**Presentation:** 2
**Contribution:** 2
**Rating:** 5
**Confidence:** 4

**Summary:**

This paper proposes RLSF, a fine-tuning approach that uses symbolic tools to provide token-level feedback for LLMs. The authors evaluate RLSF on five tasks across three domains: code generation, chemistry, and math problem solving (Game of 24).

**Strengths:**

# Implementation Details
- Clear description of the RLSF approach
- Good reproducibility through detailed experimental setup
- Comprehensive evaluation across multiple domains
# Results
- Shows consistent improvements over baselines across different tasks
- Provides detailed metrics and comparisons
- Demonstrates potential practical utility

**Weaknesses:**

# Limited Technical Novelty
- The core idea of using symbolic tools for feedback is not new - the paper acknowledges similar approaches in code generation and verification
- The main contribution appears to be applying token-level feedback in RL fine-tuning, which is an incremental advance
- The approach is essentially a straightforward combination of existing techniques (RL, symbolic verification, token-level feedback)
# Experimental Focus
- The paper is primarily focused on empirical results across three domains
- Limited theoretical analysis or justification for why this approach works better
- No significant algorithmic innovations beyond combining known components

**Questions:**

1. How does the approach scale to more complex tasks requiring deeper reasoning?
2. Why focus on these particular domains/tasks? How generalizable is the approach?

---

> ### Author Response · Authors · 2024-11-18
>
> Thank you for your feedback. We appreciate your recognition of the clarity of our implementation, the comprehensive evaluation, and the practical utility of our results. Below, we address your concerns regarding novelty, experimental focus, and scalability/generalizability.
>
> **Q1. Novelty**
>
> **A1.** We respectfully disagree with the characterization of our work as a "straightforward combination." The integration of token-level feedback via symbolic reasoning using poly-sized certificates represents a significant advance over existing approaches. Developing RLSF required significant effort in designing token-level reward functions across these tasks and adapting existing LLM fine-tuning libraries to handle fine-grained feedback from symbolic reasoning tools while keeping the framework modular. The fine-grained symbolic feedback allows precise error correction while ensuring generalizability across diverse tasks. Moreover, our work introduces several key innovations:
>
> 1. Fine-grained token-level symbolic feedback in LLMs: Unlike existing approaches that rely on scalar or binary signals [3,4,5], RLSF provides fine-grained, token-level feedback derived from poly-sized certificates generated by symbolic tools. This enables more precise error identification and correction, leading to significantly improved fine-tuning performance.
>
> 2. No need for differentiable symbolic feedback: Neurosymbolic Reinforcement Learning (NRL) approaches [1,2] typically require differentiable reasoning systems, limiting their flexibility. RLSF, by contrast, integrates non-differentiable symbolic tools into the environment, broadening its applicability while providing modular and interpretable feedback. The RLSF approach allows any LLM to be used as the agent and any symbolic tool as the environment, enabling fine-tuning of the LLM for specific tasks without requiring modifications to its architecture - unlike NRL, which necessitates architectural changes to accommodate symbolic tools.
>
> 3. Use in fine-tuning rather than in-context: The RLSF framework incorporates symbolic reasoning systems during fine-tuning, providing corrective feedback to the LLM. This approach contrasts with prior research [12,13], which only use symbolic tools during inference. By using reasoning systems during fine-tuning, RLSF enhances learning efficacy, enabling more granular adjustments to the model for the particular task.
>
> 4. Sound corrective feedback via symbolic systems: The reliance on symbolic reasoning tools ensures sound feedback, avoiding the pitfalls of self-verification or in-context feedback loops used by other LLM-based systems [6,7,8,9]. Unlike these approaches, RLSF addresses the limitations of LLMs in self-critiquing, as observed by some recent work [10,11], by relying on external symbolic tools (e.g., compilers, CAS systems), ensuring more accurate guidance.
>
> 5. Smaller and efficient models: By using symbolic feedback, RLSF significantly improves the performance of smaller models compared to closed-source, larger-scale models like GPT-3.5 and GPT-4 across five different tasks. This efficiency advantage is crucial for tasks requiring richer domain-specific understanding.
>
> **Q2. Experimental focus & generalizability**
>
> **A2.** RLSF is highly generalizable. It can scale to complex tasks as long as symbolic reasoning tools capable of verifying task-specific outputs are available. For instance, we have already started exploring RLSF-like techniques for proof synthesis, code repair, symbolic regression, material science, and physics simulators that could extend their applicability to deeper reasoning tasks. This scalability is inherent to the modular design of RLSF that ensures that it can be applied to any task with outputs expressible in a formal language, and availability of symbolic tools to provide fine-grained feedback. We discuss this in more detail at the end of Section 3 of the paper. The diverse set of five tasks we evaluated (e.g., code generation, chemistry, mathematics) underscores this generalizability.
>
> Our work emphasizes empirical validation to demonstrate the practical utility of RLSF. The approach was rigorously evaluated across five different tasks encompassing three distinct domains. Code generation is a well-studied task in symbolic reasoning, showing how RLSF improves both compilation and functional correctness metrics. Chemistry is a challenging domain where outputs must adhere to strict syntax and semantic rules, demonstrating the versatility of RLSF. Mathematical Game of 24 reasoning task highlights the benefits of symbolic feedback in guiding LLM-generated arithmetic solutions. We selected these domains because they are benchmarks for tasks with formal outputs requiring adherence to domain-specific constraints. By demonstrating improvements across these diverse domains, we illustrate RLSF’s generalizability and robustness.

---

> ### Author Response · Authors · 2024-11-18
>
> **Q3. Lack of theoretical analysis**
>
> **A3.** Token-level feedback via symbolic reasoning provides finer-grained sound reward signals than scalar model-generated feedback, aligning model updates more effectively with specific errors. This results in faster convergence and improved fine-tuning performance. Algorithm 1 explicitly details how symbolic tools generate token-level corrections, highlighting the modular and interpretable nature of RLSF.
>
> Additionally, we mention the lack of deeper theoretical guarantees in the "Limitations" section, noting that this is an area for future work. As RLSF is the first paradigm to integrate token-level feedback with symbolic tools in this manner, we consider our work an important stepping stone, providing empirical evidence to guide subsequent theoretical advancements. The demonstrated empirical performance across domains and consistent improvements over larger models such as GPT-4 highlight the substantial contribution of RLSF.
>
> We hope that the clarification of our technical contributions, the rationale for the chosen domains, and the generalizability of RLSF address your concerns. Our contributions provide significant advancements and offer a stepping stone for future research in integrating symbolic reasoning with LLM fine-tuning.
>
> Thank you again for your valuable feedback, and we respectfully request you to consider increasing your score if the clarifications address your concerns.
>
> **References:**
>
> [1] Pascal Hitzler, Md Kamruzzaman Sarker, and Aaron Eberhart. Compendium of Neurosymbolic Artificial Intelligence, volume 369. IOS Press, 2023.
>
> [2] Kamal Acharya, Waleed Raza, Carlos Dourado, Alvaro Velasquez, and Houbing Herbert Song. Neurosymbolic reinforcement learning and planning: A survey. IEEE Transactions on Artificial Intelligence, 2023.
>
> [3] Yanju Chen, Chenglong Wang, Osbert Bastani, Isil Dillig, and Yu Feng. Program synthesis using deduction-guided reinforcement learning. In International Conference on Computer Aided Verification, pp. 587–610. Springer, 2020.
>
> [4] Jia Zou, Xiaokai Zhang, Yiming He, Na Zhu, and Tuo Leng. Fgeo-drl: Deductive reasoning for geometric problems through deep reinforcement learning. Symmetry, 16(4):437, 2024.
>
> [5] Prithwish Jana, Piyush Jha, Haoyang Ju, Gautham Kishore, Aryan Mahajan, and Vijay Ganesh. Cotran: An llm-based code translator using reinforcement learning with feedback from compiler and symbolic execution. arXiv preprint arXiv:2306.06755, 2023.
>
> [6] Noah Shinn, Federico Cassano, Ashwin Gopinath, Karthik Narasimhan, and Shunyu Yao. Reflexion: Language agents with verbal reinforcement learning. Advances in Neural Information Processing Systems, 36, 2024.
>
> [7] Aman Madaan, Niket Tandon, Prakhar Gupta, Skyler Hallinan, Luyu Gao, Sarah Wiegreffe, Uri Alon, Nouha Dziri, Shrimai Prabhumoye, Yiming Yang, et al. Self-refine: Iterative refinement with self-feedback. Advances in Neural Information Processing Systems, 36, 2024.
>
> [8] Debjit Paul, Mete Ismayilzada, Maxime Peyrard, Beatriz Borges, Antoine Bosselut, Robert West, and Boi Faltings. Refiner: Reasoning feedback on intermediate representations. arXiv preprint arXiv:2304.01904, 2023.
>
> [9]Geunwoo Kim, Pierre Baldi, and Stephen McAleer. Language models can solve computer tasks. Advances in Neural Information Processing Systems, 36, 2024.
>
> [10] Yujun Mao, Yoon Kim, and Yilun Zhou. Champ: A competition-level dataset for fine-grained analyses of llms’ mathematical reasoning capabilities. arXiv preprint arXiv:2401.06961, 2024.
>
> [11] Kaya Stechly, Karthik Valmeekam, and Subbarao Kambhampati. On the self-verification limitations of large language models on reasoning and planning tasks. arXiv preprint arXiv:2402.08115, 2024.
>
> [12] Lakshya A Agrawal, Aditya Kanade, Navin Goyal, Shuvendu Lahiri, and Sriram Rajamani. Monitor-guided decoding of code lms with static analysis of repository context. Advances in Neural Information Processing Systems, 36, 2024.
>
> [13] Subbarao Kambhampati, Karthik Valmeekam, Lin Guan, Kaya Stechly, Mudit Verma, Siddhant Bhambri, Lucas Saldyt, and Anil Murthy. Llms can’t plan, but can help planning in llm-modulo frameworks. arXiv preprint arXiv:2402.01817, 2024.

---

> > ### Author Response · Authors · 2024-11-22
> >
> > Dear Reviewer,
> >
> > If there are any remaining questions or points of clarification needed after our rebuttal, we would be happy to address them. We kindly remind you to share any further thoughts or responses so that we can ensure all concerns are fully addressed. Thank you for your time and consideration.

---

### Author Response · Authors · 2024-11-24
**Revised PDF addressing reviewer feedback**

We thank the reviewers for their valuable feedback, which has helped us improve the clarity and presentation of our work. Based on the suggestions, we have made the following revisions:

1. We have updated Sections 5.1 and 6.1 and added Appendix C.2 to provide clearer explanations, along with detailed examples, of how symbolic feedback is converted into fine-grained signals for the chemistry tasks and the Game of 24.

2. We have updated Section 4.4 and added Appendix A.2 to include experiments comparing the use of binary success reward and binary compilability reward in pseudo-code to code translation task.

3. Following reviewer feedback, we have made the abstract more concise while retaining all critical elements of our contributions.

We believe these changes address the reviewers' concerns and further strengthen the manuscript. Thank you for your insightful comments and suggestions.

---

### Meta-Review · Area_Chair_a2WB · 2024-12-15

**Metareview:**

Authors present a novel fine-tuning paradigm for large language models through symbolic feedback, by providing feedback in the form of poly-sized certificates at the token level. They evaluate on five tasks across three domains: code generation, chemistry, and math problem solving (Game of 24).

Reviewers note several technical strengths of the work, including that it does not need a differentiable symbolic reasoning system, exhibits enhanced performance across multiple tasks, and the writing is clear. However, there are concerns about the novelty of the contribution. It is already widely known that fine-grained feedback outperforms sparse, response-level feedback, and reviewers note that the paper is structured as if that is the main contribution, rather than the incorporation of symbolic language for feedback.

After reading through Section 3 of the paper, I agree with reviewers that the way the paper is currently structured is not sufficient to highlight how symbolic feedback can be turned into dense reward in a general way that characterizes a meaningful contribution. I encourage the authors to revise and resubmit their manuscript to highlight this symbolic feedback as reward framework.

**Additional Comments On Reviewer Discussion:**

Reviewers ultimately decided that this was a borderline paper, where the main component was leveraging symbolic feedback to make the feedback more dense was ultimately not substantive enough to merit acceptance.

---

### Decision · Program_Chairs · 2025-01-22

Reject